# The role of structural pleiotropy and regulatory evolution in the retention of heteromers of paralogs

Axelle Marchant[1,2,3,4†], Angel F Cisneros[1,2,3†], Alexandre K Dubé[1,2,3,4], Isabelle Gagnon-Arsenault[1,2,3,4], Diana Ascencio[1,2,3,4], Honey Jain[1,2,3,5], Simon Aubé[1,2,3], Chris Eberlein[2,3,4], Daniel Evans-Yamamoto[6,7,8], Nozomu Yachie[6,7,8,9], Christian R Landry[1,2,3,4]*

[1]Département de biochimie, de microbiologie et de bio-informatique, Université Laval, Québec, Canada; [2]PROTEO, le réseau québécois de recherche sur la fonction, la structure et l'ingénierie des protéines, Université Laval, Québec, Canada; [3]Centre de Recherche en Données Massives (CRDM), Université Laval, Québec, Canada; [4]Département de biologie, Université Laval, Québec, Canada; [5]Department of Biological Sciences, Birla Institute of Technology and Sciences, Pilani, India; [6]Research Center for Advanced Science and Technology, University of Tokyo, Tokyo, Japan; [7]Institute for Advanced Biosciences, Keio University, Tsuruoka, Japan; [8]Graduate School of Media and Governance, Keio University, Fujisawa, Japan; [9]Department of Biological Sciences, Graduate School of Science, University of Tokyo, Tokyo, Japan

*For correspondence:
christian.landry@bio.ulaval.ca

†These authors contributed equally to this work

**Abstract** Gene duplication is a driver of the evolution of new functions. The duplication of genes encoding homomeric proteins leads to the formation of homomers and heteromers of paralogs, creating new complexes after a single duplication event. The loss of these heteromers may be required for the two paralogs to evolve independent functions. Using yeast as a model, we find that heteromerization is frequent among duplicated homomers and correlates with functional similarity between paralogs. Using *in silico* evolution, we show that for homomers and heteromers sharing binding interfaces, mutations in one paralog can have structural pleiotropic effects on both interactions, resulting in highly correlated responses of the complexes to selection. Therefore, heteromerization could be preserved indirectly due to selection for the maintenance of homomers, thus slowing down functional divergence between paralogs. We suggest that paralogs can overcome the obstacle of structural pleiotropy by regulatory evolution at the transcriptional and post-translational levels.

DOI: https://doi.org/10.7554/eLife.46754.001

## Introduction

Proteins assemble into molecular complexes that perform and regulate structural, metabolic and signaling functions (*Janin et al., 2008*; *Marsh and Teichmann, 2015*; *Pandey et al., 2017*; *Scott and Pawson, 2009*; *Vidal et al., 2011*; *Wan et al., 2015*). The assembly of complexes is necessary for protein function and thus constrains the sequence space available for protein evolution. One direct consequence of protein-protein interactions (PPIs) is that a mutation in a given gene can have pleiotropic effects on other genes' functions through physical associations. Therefore, to understand how genes and cellular systems evolve, we need to consider physical interactions as part of the environmental factors shaping a gene's evolutionary trajectory (*Landry et al., 2013*; *Levy et al., 2012*).

**Figure 1.** Mutations in paralogous proteins originating from an ancestral homomer are likely to have pleiotropic effects on each other's function due to their physical association. Gene duplication leads to physically interacting paralogs when they derive from an ancestral homomeric protein. The evolutionary fates of the physically associated paralogs tend to be interdependent because mutations in one gene can impact on the function of the other copy through heteromerization.
DOI: https://doi.org/10.7554/eLife.46754.002

A context in which PPIs and pleiotropy may be particularly important is during the evolution of new genes after duplication events (*Amoutzias et al., 2008*; *Baker et al., 2013*; *Diss et al., 2017*; *Kaltenegger and Ober, 2015*). The molecular environment of a protein in this context includes its paralog if the duplicates derived from an ancestral gene encoding a self-interacting protein (homomer) (*Figure 1*). In this case, mutations in one paralog could have functional consequences for the other copy because the duplication of a homomeric protein leads not only to the formation of two homomers but also to a new heteromer (*Figure 1*) (*Pereira-Leal et al., 2007*; *Wagner, 2003*). We refer to these complexes as homomers (HMs) and heteromers of paralogs (HETs).

Paralogs originating from HMs are physically associated as HETs when they arise. Subsequent evolution can lead to the maintenance or the loss of these HETs. Consequently, paralogs that maintained the ability to form HETs have often evolved new functional relationships (*Amoutzias et al., 2008*; *Baker et al., 2013*; *Kaltenegger and Ober, 2015*). Examples include a paralog degenerating and becoming a repressor of the other copy (*Bridgham et al., 2008*), pairs of paralogs that split the functions of the ancestral HM between one of the HMs and the HET (*Baker et al., 2013*), that cross-stabilize and that thus need each other to perform their function (*Diss et al., 2017*), or that evolved a new function together as a HET (*Boncoeur et al., 2012*). However, there are also paralogs that do form HMs but that have lost the ability to form HETs through evolution. Among these are duplicated histidine kinases (*Ashenberg et al., 2011*) and many heat-shock proteins (*Hochberg et al., 2018*). For the majority of HETs, we do not know what novel functions, if any, contribute to their maintenance.

Therefore, one important question to examine is: what are the evolutionary forces at work for the maintenance or the disruption of HETs arising from HMs? Previous studies suggest that if a paralog pair maintains its ability to form HMs, it is very likely to maintain the HET complex as well (*Pereira-*

*Leal et al., 2007*). For instance, *Lukatsky et al. (2007)* showed that proteins tend to intrinsically interact with themselves and that negative selection may be needed to disrupt HMs. Since nascent paralogs are identical just after duplication, they would tend to maintain a high propensity to assemble with each other. Hence, the two paralogs would form both HMs and HETs until the emergence of mutations that specifically destabilize one or the other (*Ashenberg et al., 2011*; *Hochberg et al., 2018*). In addition, the rate at which the HET is lost may depend on epistasis since it may cause mutations to be more or less disruptive together for the HET than they are individually for the HMs (*Diss and Lehner, 2018*; *Starr and Thornton, 2016*). Here, we hypothesize that the association of paralogs forming HETs acts as a constraint that may slow down the functional divergence of paralogs by making mutations on one paralog affect the function of the other.

Previous studies have shown that HMs are enriched in eukaryotic PPI networks (*Lynch, 2012*; *Pereira-Leal et al., 2007*). However, the extent to which paralogs interact with each other has not been comprehensively quantified in any species. We therefore analyze the physical assembly of HETs exhaustively in a eukaryotic interactome by integrating data from the literature and by performing a large-scale PPI screening experiment. Then, using functional data analysis, we examine the consequences of losing HET formation for paralogs forming HMs. We perform *in silico* evolution experiments to study whether the molecular pleiotropy of mutations, caused by shared binding interfaces between HM and HET complexes, could contribute to maintain interactions between paralogs originating from ancestral HMs. We show that selection to maintain HMs alone may be sufficient to prevent the loss of HETs. Finally, we find that regulatory evolution, either at the level of gene transcription or protein localization, may relieve the pleiotropic constraints maintaining the interaction of paralogous proteins.

## Results

### Homomers among singletons and paralogs in the yeast PPI network

We first examined the extent of homomerization across the yeast proteome (see dataset in Materials and methods and the supplementary text) for two classes of paralogs, those that are small-scale duplicates (SSDs) and those that are whole-genome duplicates (WGDs). We considered these two sets separately because they may have been retained through different mechanisms (see below). The dataset for this analysis, which includes previously reported PPIs and novel DHFR Protein-fragment Complementation Assay experiments (referred to as PCA, see Materials and methods and supplementary text), covers 2521 singletons, 2547 SSDs, 866 WGDs and 136 genes that are both SSDs and WGDs (henceforth referred to as 2D) (*Supplementary file 2* Tables S1 and S2). We find that among the 6070 tested yeast proteins, 1944 (32%) form HMs, which agrees with previous estimates from crystal structures (*Lynch, 2012*). The proportion of HMs among singletons (n = 630, 25%) is lower than for all duplicates: SSDs (n = 980, 38%, p-value<2.0e-16), WGDs (n = 283, 33%, p-value=1.6e-05) and 2D (n = 51, 38%, p-value=1.7e-03) (*Figure 2A Supplementary file 2* Tables S1 and S2).

Although a large number of PPIs have been previously reported in *Saccharomyces cerevisiae*, it is possible that the frequency of HMs is slightly underestimated because they were not systematically and comprehensively tested (see Materials and methods). Another reason could be that some interactions were not detected due to low expression levels. We measured mRNA abundance in cells grown in PCA conditions and used available yeast protein abundance data (*Wang et al., 2012*) to test this possibility (*Supplementary file 2* Tables S3, S4, S5 and S6). As previously observed (*Celaj et al., 2017*; *Freschi et al., 2013*), we found a correlation between PCA signal and expression level, both at the level of mRNA and protein abundance (Spearman's r = 0.33, p-value=3.5e-13 and Spearman's r = 0.46, p-value<2.2e-16 respectively). When focusing only on previously reported HMs, we also observed both correlations (Spearman's r = 0.37, p-value=3.9e-08 and Spearman's r = 0.38, p-value=6.0e-08 respectively). The association between PCA signal and expression translates into a roughly two-fold increase in the probability of HM detection when mRNA levels change by one order of magnitude (*Figure 2—figure supplement 1A*). We also generally detected stronger PCA signal for the HM of the most expressed paralog of a pair, confirming the effect of expression on our ability to detect PPIs (*Figure 2—figure supplement 1B*). Finally, we found that HMs reported in the literature but not detected by PCA have on average lower expression levels

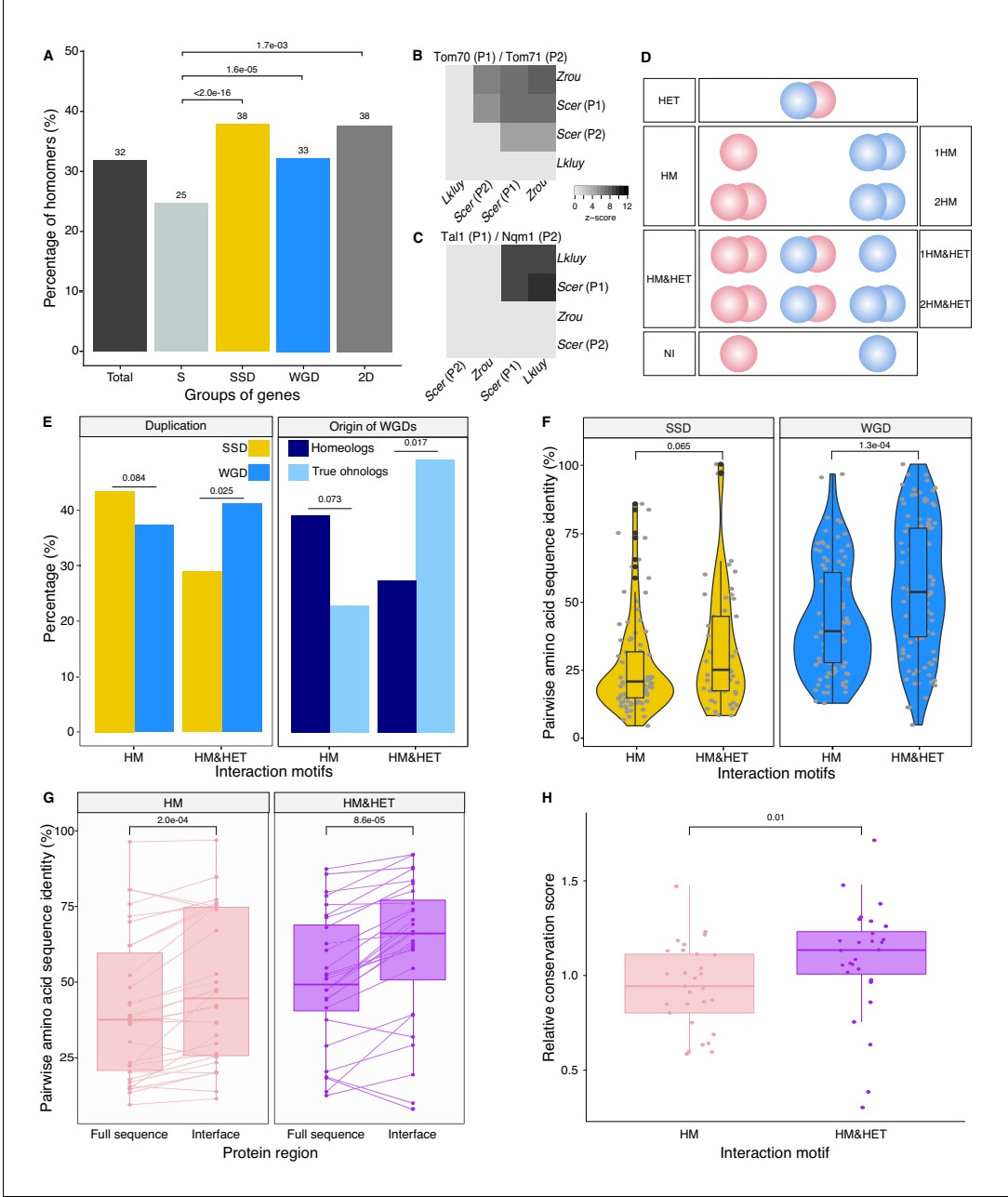

**Figure 2.** Homomers and heteromers of paralogs are frequent in the yeast protein interaction network. (**A**) The percentage of homomeric proteins in *S. cerevisiae* varies among singletons (S, n = 2521 tested), small-scale duplicates (SSDs, n = 2547 tested), whole-genome duplicates (WGDs, n = 866 tested) and genes duplicated by the two types of duplication (2D, n = 136 tested) (global Chi-square test: p-value<2.2e-16). Each category is compared with the singletons using a Fisher's exact test. P-values are reported on the graph. (**B and C**) Interactions between *S. cerevisiae* paralogs and pre-whole-genome duplication orthologs using DHFR PCA. The gray tone shows the PCA signal intensity converted to z-scores. Experiments were performed in *S. cerevisiae*. Interactions are tested among: (**B**) *S. cerevisiae* (*Scer*) paralogs Tom70 (**P1**) and Tom71 (**P2**) and their orthologs in *Lachancea kluyveri* (*Lkluy*, SAKL0E10956g) and in *Zygosaccharomyces rouxii* (*Zrou*, ZYRO0G06512g) and (**C**) *S. cerevisiae* paralogs Tal1 (**P1**) and Nqm1 (**P2**) and their orthologs in *L. kluyveri* (*Lkluy*, SAKL0B04642g) and in *Z. rouxii* (*Zrou*, ZYRO0A12914g). (**D**) Paralogs show six interaction motifs that we grouped in four categories according to their patterns. HET pairs show heteromers only. HM pairs show at least one homomer (one for 1HM or two for 2HM). HM&HET pairs show at least one homomer (one for 1HM&HET or two for 2HM&HET) and the heteromer. NI (non-interacting) pairs show no interaction. We focused our analysis on pairs derived from an ancestral HM, which we assume are pairs showing the HM and HM&HET motifs. (**E**) Percentage of HM and HM&HET among SSDs (202 pairs considered,

*Figure 2 continued on next page*

*Figure 2 continued*

yellow) and WGDs (260 pairs considered, blue) (left panel), homeologs that originated from inter-species hybridization (47 pairs annotated and considered, dark blue) (right panel) and true ohnologs from the whole-genome duplication (82 pairs annotated and considered, light blue). P-values are from Fisher's exact tests. (**F**) Percentage of pairwise amino acid sequence identity between paralogs for HM and HM&HET motifs for SSDs and WGDs. P-values are from Wilcoxon tests. (**G**) Pairwise amino acid sequence identity for the full sequences of paralogs and their binding interfaces for the two motifs HM and HM&HET. P-values are from paired Wilcoxon tests. (**H**) Relative conservation scores for the two motifs of paralogs. Conservation scores are the percentage of sequence identity at the binding interface divided by the percentage of sequence identity outside the interface. Data shown include 30 interfaces for the HM group and 28 interfaces for the HM&HET group (22 homomers and 3 heterodimers of paralogs) (*Supplementary file 2* Table S13). P-value is from a Wilcoxon test.
DOI: https://doi.org/10.7554/eLife.46754.003

The following figure supplements are available for figure 2:

**Figure supplement 1.** Association between mRNA abundance and the probability of HM detection by PCA in this study.
DOI: https://doi.org/10.7554/eLife.46754.006

**Figure supplement 2.** mRNA and protein abundance of singletons and duplicates.
DOI: https://doi.org/10.7554/eLife.46754.007

**Figure supplement 3.** Comparison of PCA data generated in this study with published data.
DOI: https://doi.org/10.7554/eLife.46754.004

**Figure supplement 4.** Intersections of detected HMs.
DOI: https://doi.org/10.7554/eLife.46754.005

**Figure supplement 5.** Interaction motifs and percentage of pairwise amino acid sequence identity between paralogs.
DOI: https://doi.org/10.7554/eLife.46754.008

**Figure supplement 6.** Conservation of binding interfaces of human paralogs in HM&HET complexes with solved structures.
DOI: https://doi.org/10.7554/eLife.46754.009

**Figure supplement 7.** Plate organization for DHFR PCA experiments.
DOI: https://doi.org/10.7554/eLife.46754.010

**Figure supplement 8.** Density of colony size converted to z-score.
DOI: https://doi.org/10.7554/eLife.46754.011

(*Figure 2—figure supplement 1B-C*). We therefore conclude that some HMs (and also HETs) remain undetected because of low expression levels.

The overrepresentation of HMs among duplicates was initially observed for human paralogs (*Pérez-Bercoff et al., 2010*). One potential mechanism to explain this finding is that homomeric proteins are more likely to be maintained as pairs after duplication because they might become dependent on each other for their stability that is enhanced through the formation of HET (*Diss et al., 2017*). Another explanation is that proteins forming HMs could be expressed at higher levels and thus more easily detected, as shown above. High expression levels are also associated with a greater long term probability of genes to persist after duplication (*Paramecium Post-Genomics Consortium et al., 2010*; *Gout and Lynch, 2015*). We indeed observed that both SSDs and WGDs are more expressed than singletons at the mRNA and protein levels, with WGDs being more expressed than SSDs at the mRNA level (*Figure 2—figure supplement 2A-B*). However, expression level (and thus PPI detectability) does not explain completely the enrichment of HMs among duplicated proteins. Both factors, expression and duplication, have significant effects on the probability of proteins to form HMs (*Supplementary file 2* Table S7. A). It is therefore likely that the overrepresentation of HMs among paralogs is linked to their higher expression along with other factors.

## Paralogs that form heteromers tend to have higher sequence identity

The model presented in *Figure 1* assumes that the ancestral protein leading to HET formed a HM before duplication. Under the principle of parsimony, we can assume that when at least one paralog forms a HM, the ancestral protein was also a HM. This was shown to be true in general by *Diss et al. (2017)*, who compared yeast WGDs to their orthologs from *Schizosaccharomyces pombe*. To further support this observation, we used PCA to test for HM formation for orthologs from species that

diverged prior to the whole-genome duplication event (*Lachancea kluyveri* and *Zygosaccharomyces rouxii*). We looked at paralogs of the mitochondrial translocon complex and the transaldolase, which show HETs according to previous studies (see Materials and methods). We confirm that when one HM was observed in *S. cerevisiae*, at least one ortholog from pre-whole-genome duplication species formed a HM (*Figure 2B-C*). We also detected interactions between orthologs, suggesting that the ability to interact has been preserved despite the millions of years of evolution separating these species. The absence of interactions for some of these orthologous proteins may be due to the incompatibility of their expression in *S. cerevisiae* or the use of a non-endogenous promoter for these experiments.

We focused on HMs and HETs for 202 pairs of SSDs and 260 pairs of WGDs. This is a reduced dataset compared to the previous section because we needed to consider only pairs for which there was no missing PPI data (see Materials and methods). We combined public data with our own PCA experimental data on 86 SSDs and 149 WGDs (see supplementary text, *Figure 2—figure supplements 3,4*). Overall, the data represents a total of 462 pairs of paralogs (202 SSDs and 260 WGDs) covering 53% of the SSDs and 50% of the WGDs (*Supplementary file 2* Tables S3 and S4). This dataset encompasses 493 binary interactions of paralogs with themselves (HMs) and 214 interactions with their sister copy (HET).

We classified paralogous pairs into four classes according to whether they show only the HET (HET, 10%), at least one HM but no HET (HM, 39%), at least one of the HM and the HET (HM&HET, 37%) or no interaction (NI, 15%) (*Figure 2D*, supplementary text). Overall, most pairs forming HETs also form at least one HM (79%, *Supplementary file 2* Table S3). For the rest of the study, we focused our analysis and comparisons on HM and HM&HET pairs because they most likely derive from an ancestral HM. Previous observations showed that paralogs are enriched in protein complexes comprising more than two distinct subunits, partly because these complexes evolved by the initial establishment of self-interactions followed by the duplication of the homomeric proteins (*Musso et al., 2007*; *Pereira-Leal et al., 2007*). However, we find that the majority of HM&HET pairs could be simple oligomers of paralogs that do not involve other proteins and are thus not part of large complexes. Only 70 (41%) of the 169 cases of HM&HET are in complexes with more than two distinct subunits among a set of 5535 complexes reported in databases (see Materials and methods).

We observed that the correlation between HM and HET formation is affected by whether paralogs are SSDs or WGDs (*Figure 2E*). WGDs tend to form HETs more often when they form at least one HM, resulting in a larger proportion of HM&HET motifs than SSDs. We hypothesize that since SSDs have appeared at different evolutionary times, many of them could be older than WGDs, which could be accompanied by a loss of interactions between paralogs. Indeed, we observed that the distribution of sequence divergence shows lower identity for SSDs than for WGDs, suggesting the presence of ancient duplicates that predate the whole-genome duplication (*Figure 2—figure supplement 5A*). Higher protein sequence divergence could lead to the loss of HET complexes because it increases the probability of divergence at the binding interface. We indeed found that among SSDs, those forming HM&HET tend to show a marginally higher overall sequence identity (p=0.065, *Figure 2F*, *Figure 2—figure supplement 5B and C*). We also observed a significantly higher sequence identity for WGD pairs forming HM&HET, albeit with a wider distribution (*Figure 2F*, *Figure 2—figure supplement 5C*). This wider distribution derives at least partly from the mixed origin of WGDs (*Figure 2—figure supplement 5D and E*). A recent study (*Marcet-Houben and Gabaldón, 2015*; *Wolfe, 2015*) showed that WGDs likely have two distinct origins: actual duplication (generating true ohnologs) and hybridization between species (generating homeologs). For pairs whose ancestral state was a HM, we observed that true ohnologs have a tendency to form HET more frequently than homeologs (*Figure 2E*). Because homeologs had already diverged before the hybridization event, they are older than ohnologs, as shown by their lower pairwise sequence identity (*Figure 2—figure supplement 5D*). This observation supports the fact that younger paralogs derived from HMs are more likely to form HETs than older ones.

Amino acid sequence conservation could also have a direct effect on the retention of HETs, independently of the age of the duplication. For instance, among WGDs (either within true ohnologs or homeologs), which all have the same age in their own category, HM&HET pairs have higher sequence identity than HM pairs (*Figure 2—figure supplement 5B, C and E*). This is also apparent for pairs of paralogs whose HM or HET structures have been solved by

crystallography (n = 58 interfaces) (*Supplementary file 2* Table S3). Indeed, we found that pairwise amino acid sequence identity was higher for HM&HET than for HM pairs for both entire proteins and for their binding interfaces (*Figure 2G*). Furthermore, the conservation ratio of the binding interface to the non-interface regions within the available structures is higher for those forming HM&HET, suggesting a causal link between sequence identity at the interface and assembly of HM and HETs (*Figure 2H*). We extended these analyses to a dataset of human paralogs (*Lan and Pritchard, 2016*; *Singh et al., 2015*) to evaluate if these trends can be generalized. Whereas interfaces within PDB structures (n = 65 interfaces) are more conserved than the full sequence for both HM and HM&HET motifs (*Figure 2—figure supplement 6A*), we did not observe differences in the ratio of conservation of interfaces to non-interfaces (*Figure 2—figure supplement 6B*). The reasons for this difference between yeast and humans remain to be explored but it could be caused by mechanisms that do not depend on interfaces to separate paralogous proteins in humans, for instance tissue-specific expression.

Considering that stable interactions are often mediated by protein domains, we looked at the domain composition of paralogs using the Protein Families Database (Pfam) (*El-Gebali et al., 2019*). We tested if differences in domain composition could explain the frequency of different interaction motifs. We found that 367 of 448 pairs of paralogs (82%) shared all their domain annotations (*Supplementary file 2* Table S3). Additionally, HM&HET paralogs tend to have more domains in common but the differences are non-significant and appear to be caused by overall sequence divergence (*Figure 3—figure supplement 1A-B*). Domain gains and losses are therefore unlikely to contribute to the loss of HET complexes following the duplication of homomers.

## Heteromer formation correlates with functional conservation

To test if the retention of HETs correlates with the functional similarity of HM and HM&HET paralogs, we used the similarity of Gene Ontology (GO) terms, reported growth phenotypes of loss-of-function mutants and patterns of genome-wide genetic interactions. These features represent the relationship of genes with cell growth and the gene-gene relationships underlying cell growth. The use of GO terms could bias the analysis because they are often predicted based on sequence features. However, phenotypes and genetic interactions are derived from unbiased experiments because interactions are tested without *a priori* consideration of a protein's functions (*Costanzo et al., 2016*). We found that HM&HET pairs are more similar than HM for SSDs (*Figure 3* and *Figure 3—figure supplement 2*). We observed the same trends for WGDs, although some of the comparisons are either marginally significant or non-significant (*Figure 3*, comparison between true ohnologs and homeologs in *Figure 3—figure supplement 3*). The higher functional similarity observed for HM&HET pairs could be the result of the higher sequence identity described above. However, for a similar level of sequence identity, HM&HET pairs have higher correlation of genetic interaction profiles, higher GO molecular function (for SSDs) and higher GO biological process similarity (for both SSDs and WGDs) than HM pairs (*Figure 3—figure supplement 4* and GLM test in *Supplementary file 2* Table S7. B). Overall, the retention of HETs after the duplication of HMs appears to correlate with functional similarity, independently from sequence conservation.

## Pleiotropy contributes to the maintenance of heteromers

Since molecular interactions between paralogs predate their functional divergence, it is likely that physical association by itself affects the retention of functional similarity among paralogs. Any feature of paralogs that contributes to the maintenance of the HET state could therefore have a strong impact on the fate of new genes emerging from the duplication of HMs. A large fraction of HMs and HETs use the same binding interface (*Bergendahl and Marsh, 2017*), so mutations at the interface may have pleiotropic effects on both HMs and HETs (*Figure 1*), which would lead to correlated responses to selection. If we assume that HMs need to self-interact in order to perform their function, it is expected that natural selection would favor the maintenance of self-assembly. Negative selection on HM interfaces would act on their pleiotropic residues and thus also preserve HET interfaces, preventing the loss of HETs as a correlated response.

We tested this correlated selection model using *in silico* evolution of HM and HET protein complexes (*Figure 4A*). We used a set of six representative high-quality structures of HMs (*Dey et al., 2018*). We evolved these HM complexes by duplicating them and following the binding energies of

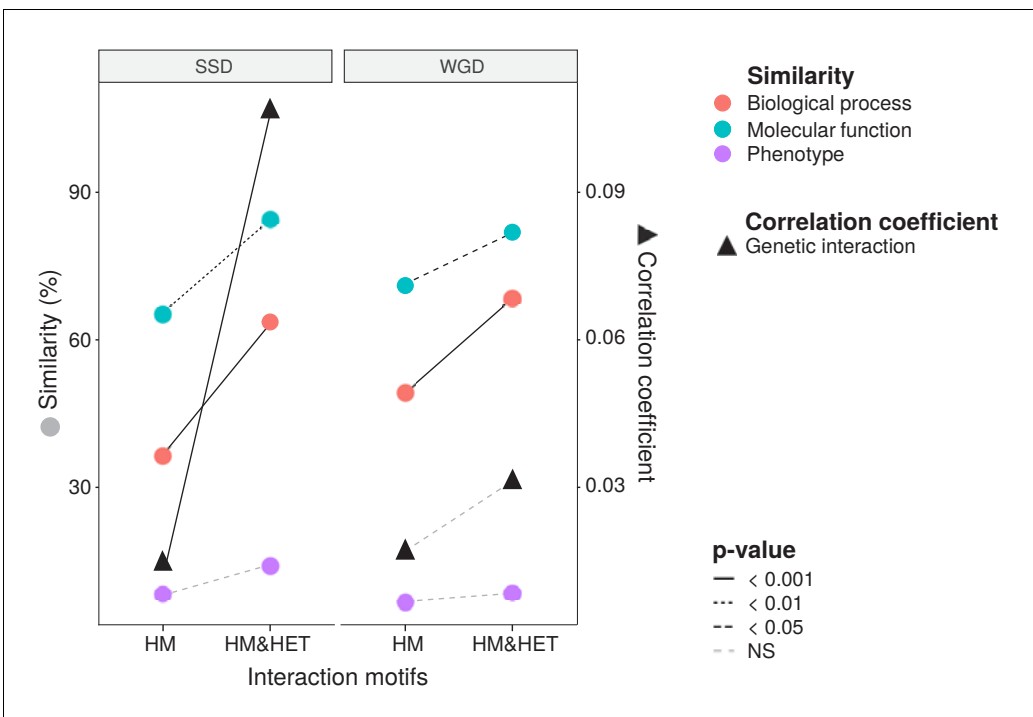

**Figure 3.** Maintenance of heteromerization between paralogs leads to greater functional similarity. The similarity score is the average proportion of shared terms (100% * Jaccard's index) across pairs of paralogs for GO molecular functions, GO biological processes and gene deletion phenotypes. The mean values of similarity scores and of the correlation of genetic interaction profiles are compared between HM and HM&HET pairs for SSDs and WGDs. P-values are from Wilcoxon tests.

DOI: https://doi.org/10.7554/eLife.46754.012

The following figure supplements are available for figure 3:

**Figure supplement 1.** Comparison of Pfam domain composition similarity between pairs of paralogs.

DOI: https://doi.org/10.7554/eLife.46754.013

**Figure supplement 2.** Comparison of functional similarity between HM and HM&HET pairs.

DOI: https://doi.org/10.7554/eLife.46754.014

**Figure supplement 3.** Comparison of functional similarity between WGDs, considering homeologs and true ohnologs separately.

DOI: https://doi.org/10.7554/eLife.46754.015

**Figure supplement 4.** Functional similarity between paralogs as a function of their pairwise amino acid sequence identity.

DOI: https://doi.org/10.7554/eLife.46754.016

the resulting two HMs and HET. We let mutations occur at the binding interface 1) in the absence of selection (neutral model), 2) in the presence of negative selection maintaining only one HM, and 3) with negative selection retaining both HMs. In these three cases, we applied no selection on binding energy of the HET. In the fourth scenario, we applied selection on the HET but not on the HMs to examine if selection maintaining the HET could also favor the retention of HMs. Mutations that have deleterious effects on the complex under selection were lost or allowed to fix with exponentially decaying probability depending on the fitness effect (see Materials and methods) (*Figure 4A*).

We find that neutral evolution leads to the destabilization of all complexes derived from the simulated duplication of a HM (PDB: 1M38) (*Figure 4B*), as is expected given that there are more destabilizing mutations than stabilizing ones (*Brender and Zhang, 2015*; *Guerois et al., 2002*). Selection to maintain one HM or both HMs significantly slows down the loss of the HET with respect to the neutral scenario (*Figure 4C-E*). Interestingly, the HET is being destabilized more slowly than the second HM when only one HM is under negative selection. The difficulty of losing the HET in the simulations could explain why for some paralog pairs, only one HM and the HET are preserved, as well as

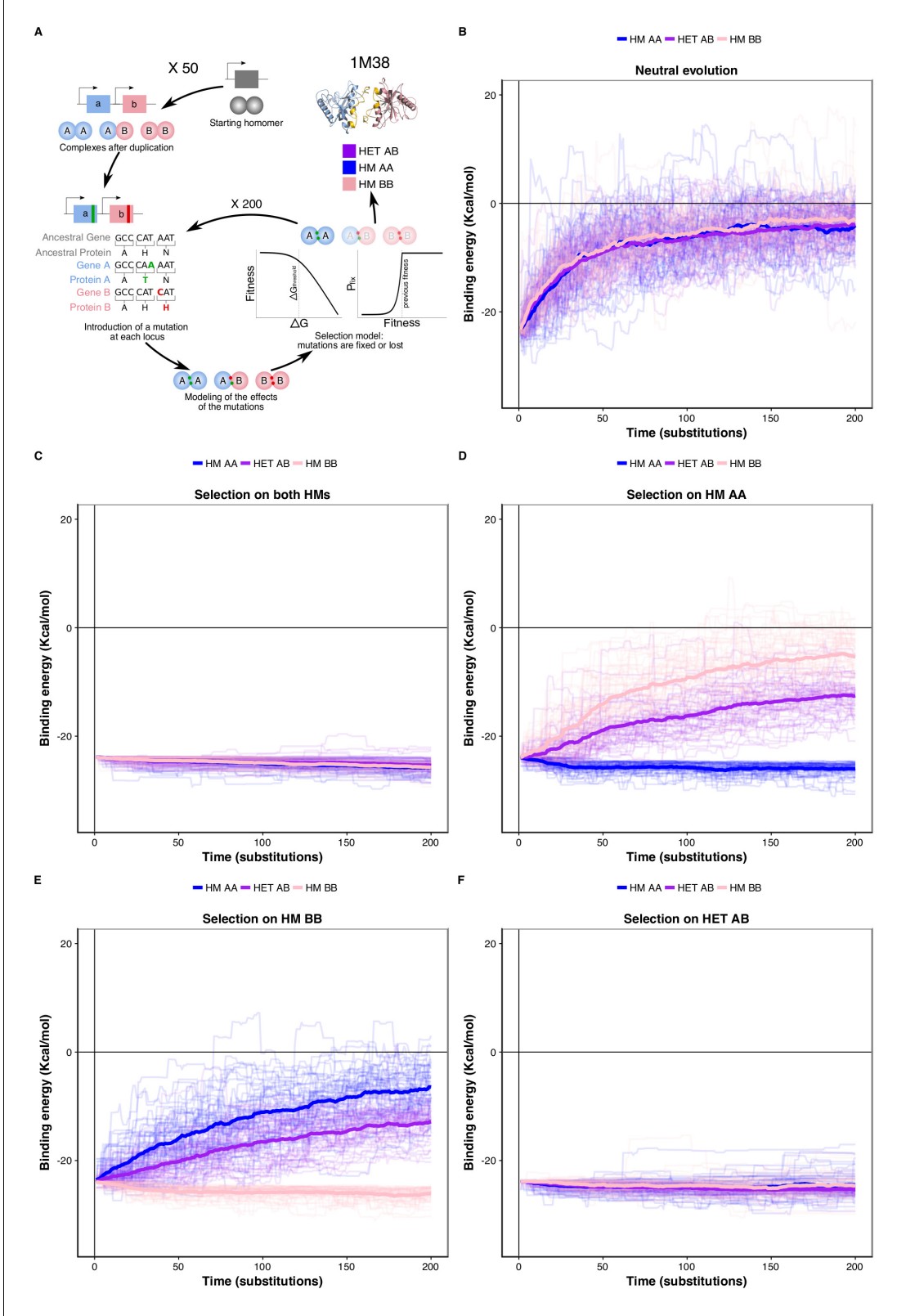

**Figure 4.** Negative selection to maintain homomers also maintains heteromers. (**A**) The duplication of a gene encoding a homomeric protein and the evolution of the complexes is simulated by applying mutations to the corresponding subunits A and B. Only mutations that would require a single nucleotide change are allowed. Stop codons are disallowed. After introducing mutations, the selection model is applied to complexes and mutations are fixed or lost. (**B to F**) The binding energy of the HMs and the HET resulting from the duplication of a HM (PDB: 1M38) is followed through time

*Figure 4 continued on next page*

*Figure 4 continued*

under different selection regimes applied on protein stability and binding energy. More positive values indicate less favorable binding and more negative values indicate more favorable binding. (B) Accumulation and neutral fixation of mutations. (C) Selection on both HMs while the HET evolves neutrally. (D) Selection on HM AA or (E) HM BB: selection maintains one HM while the HET and the other HM evolve neutrally. (F) Selection on HET while the HMs evolve neutrally. (E) Selection on HM AA or (F) HM BB: selection maintains one HM while the HET and the other HM evolve neutrally. Mean binding energies among replicates are shown in thick lines and the individual replicates are shown with thin lines. Fifty replicate populations are monitored in each case and followed for 200 substitutions. PDB structure 1M38 was visualized with PyMOL (**Schrödinger LLC, 2015**). The number of substitutions that are fixed on average during the simulations are shown in **Supplementary file 2** Table S8.

DOI: https://doi.org/10.7554/eLife.46754.017

The following figure supplements are available for figure 4:

**Figure supplement 1.** Percentage of interaction motifs for SSDs, WGDs and the two types of WGDs.

DOI: https://doi.org/10.7554/eLife.46754.018

**Figure supplement 2.** Similar evolutionary trajectories are observed for six different PDB structures.

DOI: https://doi.org/10.7554/eLife.46754.019

**Figure supplement 3.** Effect of changes in parameters on the observed evolution trajectories.

DOI: https://doi.org/10.7554/eLife.46754.020

**Figure supplement 4.** Single mutants have pleiotropic effects for HM and HET.

DOI: https://doi.org/10.7554/eLife.46754.021

why there are few pairs of paralogs that specifically lose the HET (*Figure 4—figure supplement 1*). The reciprocal situation is also true, i.e. negative selection on HET significantly decelerates the loss of stability of both HMs (*Figure 4F*). These observations hold when simulating the evolution of duplication of five other structures (*Figure 4—figure supplement 2*) and when simulating evolution under different combinations of the parameters that control the efficiency of selection and the length of the simulations (*Figure 4—figure supplement 3*). By examining the effects that single mutants (only one of the loci gets a nonsynonymous mutation) have on HMs and HET, we find that, as expected, their effects are strongly correlated and thus highly pleiotropic (Pearson's r between 0.64 and 0.9 (*Figure 4—figure supplement 4*)). We observe strong pleiotropic effects of mutations for the six structures tested, which explains the correlated responses to selection in the *in silico* evolution. Additionally, mutations tend to have greater effects on the HM than on the HET (*Figure 4—figure supplement 4*, *Figure 5—figure supplement 1*), which agrees with observations on HMs having a greater variance of binding energies than HETs (*André et al., 2008*; *Lukatsky et al., 2007*; *Lukatsky et al., 2006*). As a consequence, HMs that are not under selection in our simulations show higher variability in their binding energy than HETs.

We examined the effects of double mutants (the two loci get a non-synonymous mutation at the interface) on HET formation to study how epistasis may influence the maintenance or loss of HET and HMs when the former or the latter are under selection. We defined epistatic effects as deviations between the observed and the expected effects of mutations on binding energy. Expected effects on HETs were calculated as the average of the effects on the HMs, which each have two subunits with the same mutation. We defined positive epistasis as cases where the observed binding is stronger than expected (more negative $\Delta\Delta G$) and negative epistasis when it is weaker (more positive $\Delta\Delta G$). In terms of evolutionary responses, positive epistasis would contribute to the retention of the HET and negative epistasis to its loss.

Regardless of the selection scenario, the mutations sampled are slightly enriched for positive epistasis, since the slope values of regression models are smaller than one (0.91 and 0.89 under selection on HMs and HET respectively). When the HMs are maintained by selection, this slightly positive epistasis is also visible in the mutations that are fixed because the epistatic effects are not selected upon. This results in a similar slope for the selected mutations as for the rejected ones. Positive epistasis may therefore contribute to the maintenance of the HET (*Figure 5A*). On the other hand, selection on the HET results in a further enrichment of mutations with positive epistasis (slope = 0.51, *Figure 5B*). In this case, mutations tolerated in the HETs and thus fixed are more destabilizing to the HMs. This is also visible in the higher number of fixed substitutions (*Supplementary file 2* Table S8) when selection acts on the HET than when it acts on both HMs, particularly for mutations having opposite effects on the HMs (*Figure 5—figure supplement 2*). This is also manifested in significantly stronger positive epistasis among fixed pairs of mutations when the

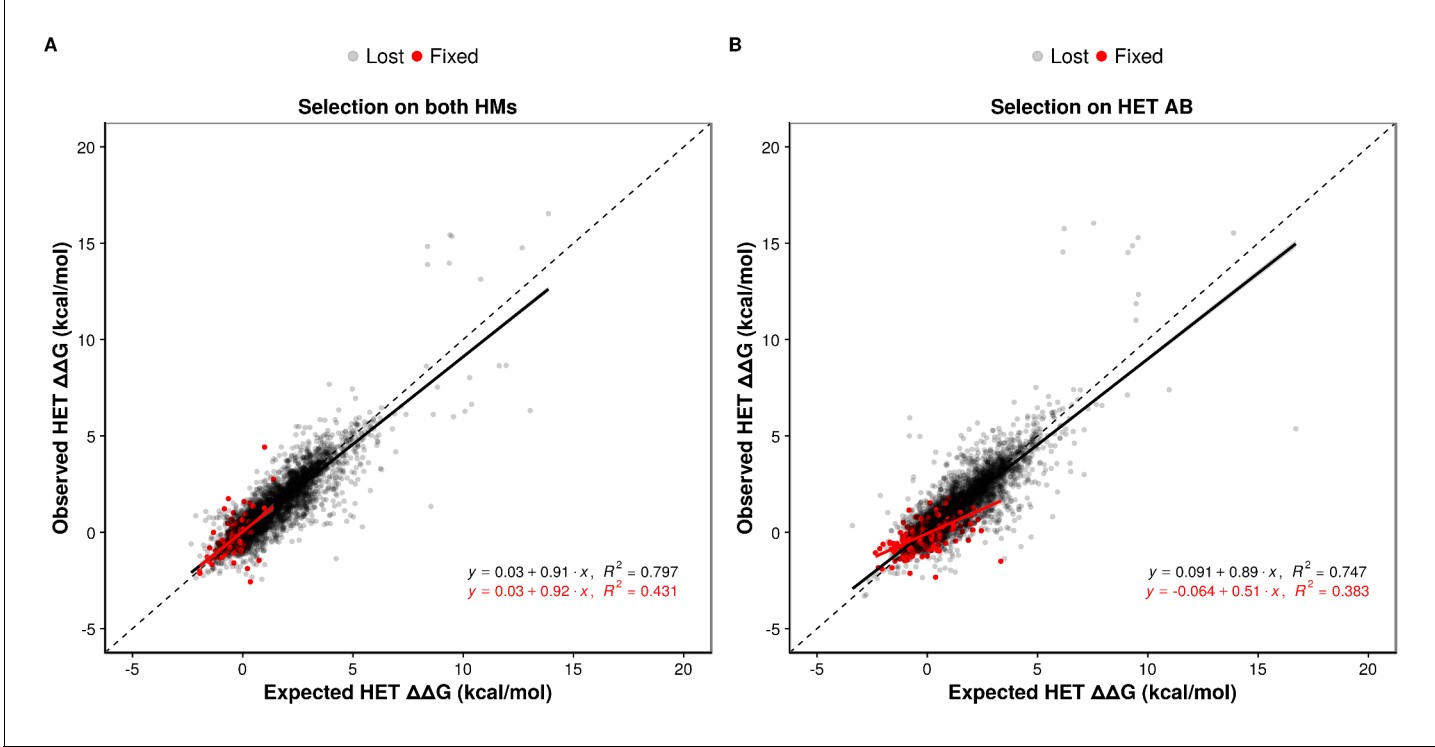

**Figure 5.** Epistasis favors the maintenance of HETs and the loss of HMs. (**A and B**) Observed effects of double mutants on HET (y-axis) are compared to their expected effects (x-axis) based on the average of their effects on the HMs when selection is applied on both HMs (n = 6777 pairs of mutations) (**A**) or on the HET (n = 6760 pairs of mutations) (**B**). Dashed lines indicate the diagonal for perfect agreement between observations and expectations (no epistasis), black regression lines indicate the best fit for the lost mutants, and red regression lines indicate the best fit for the fixed mutants. Data were obtained from simulations with PDB structure 1M38. The regression coefficients, intercepts and $R^2$ values are indicated on the figure for fixed and lost mutations. A regression coefficient lower than one means that pairs of mutations have a less destabilizing effects on the HET than expected based on their average effects on the HMs.

DOI: https://doi.org/10.7554/eLife.46754.022

The following figure supplements are available for figure 5:

**Figure supplement 1.** Distribution of effect sizes of mutations on the binding energy (ΔΔG) of HMs and HETs as estimated using FoldX.

DOI: https://doi.org/10.7554/eLife.46754.024

**Figure supplement 2.** Fixation rates of double mutants during the simulations.

DOI: https://doi.org/10.7554/eLife.46754.025

**Figure supplement 3.** Contribution of epistasis to the evolution of HET for six different PDB structures.

DOI: https://doi.org/10.7554/eLife.46754.023

---

HET is under negative selection (t-test, p-value=0.009). These observations suggest that epistasis may make HETs more robust to mutations than HMs with respect to protein complex assembly, contributing to their maintenance when the HMs are under selection and contributing to the loss of HMs when the HET is under negative selection. This effect is visible in our simulations since selection on the HET results in a slow destabilization of the two HMs (*Figure 4*, *Figure 4—figure supplement 2*), especially when more mutations are attempted (*Figure 4—figure supplement 3*), and is observed for all six structures tested (*Figure 5—figure supplement 3*).

## Regulatory evolution may break down molecular pleiotropy

The results from simulations show that the loss of HET after the duplication of a HM occurs at a slow rate if HMs are maintained by selection and that specific rare mutations may be required for HETs to be destabilized. However, the simulations only consider the evolution of binding interfaces, which limits the modification of interactions to a subset of all mutations that can ultimately affect PPIs (*Hochberg et al., 2018*). Other mechanisms that would lead to the loss of HETs could involve transcriptional regulation or cell compartment localization such that paralogs are not present at the

same time or in the same cell compartment. To test how regulatory evolution affects interactions, we measured the correlation coefficient of expression profiles of paralogs using mRNA microarray measurements across more than 1000 growth conditions (*Ihmels et al., 2004*). These expression profiles are more correlated for both SSD and WGD paralogs forming HM&HET than for those forming only HM (p-value=6.5e-03 and 6.1e-03 respectively, *Figure 6A*). This result holds using available single-cell RNAseq data (*Gasch et al., 2017*) although the trend is not significant for WGDs (*Figure 6—figure supplement 1A*). Because we found that sequence identity was correlated with both the probability of observing HM&HET and the co-expression of paralogs, we tested if co-expression had an effect on HET formation when controlling for sequence identity. For SSDs, co-expression shows significant effects on HM&HET formation (*Figure 6C*, *Figure 6—figure supplement 1B*. and *Supplementary file 2* Table S7. B) but not for WGDs (*Figure 6C*, *Figure 6—figure supplement 1B*. and *Supplementary file 2* Table S7. B). This is true also when considering the two origins of WGDs separately (*Figure 6—figure supplement 2A-F*). The differences of expression correlation between HM and HM&HET could be caused by *cis* regulatory divergence, for instance, HM&HET pairs might have more similar transcription factor binding sites. While we do observe a marginally higher transcription factor binding site similarity for HM&HET pairs than for HM pairs, the tendency is not significant, suggesting other causes for the divergence and similarity of expression profiles (*Figure 6B*, *Figure 6—figure supplement 3* and *Supplementary file 2* Table S7. B).

Finally, we find that HM&HET paralogs are more similar than HM for both SSDs and WGDs in terms of cellular compartments (GO) and cellular localization derived from experimental data (*Figure 6C*, *Figure 6—figure supplement 3B C*). For a similar level of sequence identity, HM&HET pairs have higher cellular compartment and cellular localization similarity (for both SSDs and WGDs) than HM pairs (*Figure 6—figure supplement 4* and GLM test in *Supplementary file 2* Table S7. B). The same tendencies are observed when considering the two classes of WGDs separately (*Figure 6—figure supplement 2G-I*).

Overall, coexpression, localization and GO cellular component comparison results suggest that changes in gene and protein regulation could prevent the interaction between paralogs that derive from ancestral HMs, reducing the role of structural pleiotropy in maintaining their associations.

## Discussion

Upon duplication, the properties of proteins are inherited from their ancestors, which may affect how paralogs subsequently evolve. Here, we examined the extent to which physical interactions between paralogs are preserved after the duplication of HMs and how these interactions affect functional divergence. Using reported PPI data, crystal structures and new experimental data, we found that paralogs originating from ancestral HMs are more likely to functionally diverge if they lost their ability to form HETs. We propose that non-adaptive mechanisms could play a role in the retention of physical interactions and in turn, impact on functional divergence. By developing a model of *in silico* evolution of PPIs, we found that molecular pleiotropic and epistatic effects of mutations on binding interfaces can constrain the maintenance of HET complexes even if they are not under selection. We hypothesize that this non-adaptive constraint could play a role in slowing down the divergence of paralogs but that it could be counteracted at least partly by regulatory evolution.

The proportions of HMs and HETs among yeast paralogs were first studied more than 15 years ago (*Wagner, 2003*). It was then suggested that most paralogs forming HETs do not have the ability to form HMs and thus, that evolution of new interactions was rapid. Since then, many PPI experiments have been performed (*Chatr-Aryamontri et al., 2017*; *Kim et al., 2019*; *Stark, 2006*; *Stynen et al., 2018*) and the resulting global picture is different. We found that most of the paralogs forming HETs also form HMs, suggesting that interactions between paralogs are inherited rather than gained *de novo*. This idea is supported by models predicting interaction losses to be much more likely than interaction gains after gene duplication (*Gibson and Goldberg, 2009*; *Presser et al., 2008*). Accordingly, the HM&HET state can be more readily achieved by the duplication of an ancestral HM rather than by the duplication of a monomeric protein followed by the gain of the HMs and of the HET. Interacting paralogs are therefore more likely to derive from ancestral HMs, as also shown by *Diss et al. (2017)* using limited comparative data. For two pairs of *S. cerevisiae* paralogs presenting the HM&HET motif in the litterature, we indeed detected HM formation of their orthologs from pre-whole-genome duplication species, supporting the model by which self-

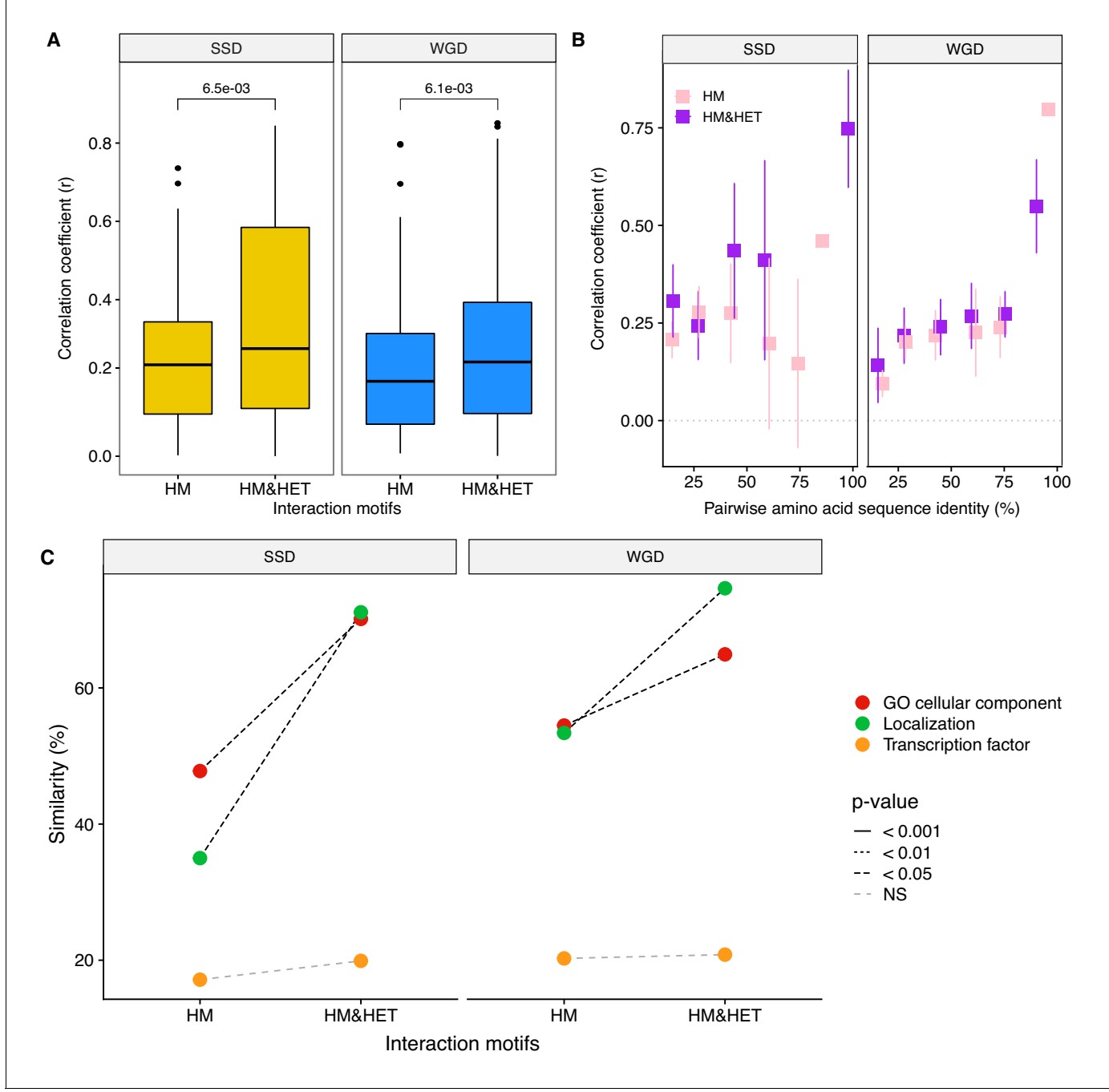

**Figure 6.** Loss of heteromerization between paralogs may result from regulatory divergence. (**A**) Correlation coefficients (Spearman's r) between the expression profiles of paralogs. The data derives from mRNA relative expression across 1000 growth conditions (***Ihmels et al., 2004***). HM and HM&HET are compared for SSDs (yellow) and WGDs (blue). P-values are from t-tests. (**B**) Correlation of expression profiles between paralogs forming only HM (pink) or HM&HET (purple) as a function of their amino acid sequence identity. The data was binned into six equal categories for representation only. (**C**) Similarity of GO cellular component, GFP-based localization, and transcription factor binding sites (100% * Jaccard's index) are compared between HM and HM and HET for SSDs and WGDs. P-values are from Wilcoxon tests.

DOI: https://doi.org/10.7554/eLife.46754.026

The following figure supplements are available for figure 6:

**Figure supplement 1.** The loss of HETs may result from regulatory divergence (single cell RNAseq data; ***Gasch et al., 2017***).
DOI: https://doi.org/10.7554/eLife.46754.027

**Figure supplement 2.** Expression of WGDs and consequences on interaction motifs.

*Figure 6 continued on next page*

*Figure 6 continued*

DOI: https://doi.org/10.7554/eLife.46754.029

**Figure supplement 3.** Interaction motifs and similarity of functions for SSDs and WGDs.

DOI: https://doi.org/10.7554/eLife.46754.028

**Figure supplement 4.** Similarity of regulation between paralogs as a function of their pairwise amino acid sequence identity.

DOI: https://doi.org/10.7554/eLife.46754.030

interactions and cross-interactions are inherited from the duplication. However, we did not detect HMs for both pre-whole-genome duplication species, which may reflect the incorrect expression of these proteins in *S. cerevisiae* rather than their lack of interaction.

We observed an enrichment of HMs among yeast duplicated proteins compared to singletons, as reported in previous studies (*Ispolatov et al., 2005*; *Pereira-Leal et al., 2007*; *Pérez-Bercoff et al., 2010*; *Yang et al., 2003*). Also, analyses of PPIs from large-scale experiments have shown that interactions between paralogous proteins are more common than expected by chance (*Ispolatov et al., 2005*; *Musso et al., 2007*; *Pereira-Leal et al., 2007*). Several adaptive hypotheses have been suggested to explain the over-representation of interacting paralogous proteins. For instance, HMs may be preferentially retained, over other duplicates, due to their capacity to provide new adaptive traits by gaining novel functions (neofunctionalization), or by splitting the original ones (subfunctionalization). Similarly, symmetrical HM proteins could have key advantages over monomeric ones for protein stability and regulation (*André et al., 2008*; *Bergendahl and Marsh, 2017*). *Levy and Teichmann (2013)* suggested that the duplication of HM proteins serves as a seed for the growth of protein complexes. These duplications would allow the diversification of complexes by the asymmetric gain or loss of interactions, which would ultimately lead to the specialization of the duplicates. It is also possible that the presence of HETs itself offers a rapid way to evolve new functions. Examples include bacterial multidrug efflux transporters (*Boncoeur et al., 2012*) and regulatory mechanisms that evolved this way (*Baker et al., 2013*; *Bridgham et al., 2008*; *De Smet et al., 2013*; *Kaltenegger and Ober, 2015*). Finally, cotranslational folding has been shown to be a problem for homomeric proteins because of premature assembly of protein complexes, particularly for proteins with interfaces closer to their N-terminus (*Natan et al., 2018*). The replacement of such HMs by HETs could solve this issue by separating the translation of the proteins to be assembled on two distinct mRNAs.

Non-adaptive mechanisms could also be at play to maintain HETs. Our simulated evolution of the duplication of HMs leads to the proposal of a simple mechanism for the maintenance of HET that does not require adaptive mechanisms. A large fraction of HMs and HETs use the same binding interface (*Bergendahl and Marsh, 2017*) and as a consequence, negative selection on HM interfaces will also preserve HET interfaces. Our results show that mutations have correlated effects on HM and HET, which slows down the divergence of these complexes. Since some proteins are unstable in the absence of their paralog and lose their capacity to interact with other proteins, cross-stabilization could be another non-adaptive mechanism for the maintenance of the HET (*Diss et al., 2017*). Notably, these proteins are enriched for paralogs forming HET, suggesting that the individual proteins depend on each other through these physical interactions (*Diss et al., 2017*). Independent observations by *DeLuna et al. (2010)* also showed that the deletion of a paralog was sometimes associated with the degradation of the sister copy, particularly among HET paralogs. The Diss et al. and DeLuna et al. observations led to the proposal that paralogs could accumulate complementary degenerative mutations at the structural level after the duplication of a HM (*Diss et al., 2017*; *Kaltenegger and Ober, 2015*). This scenario would lead to the maintenance of the HET because destabilizing mutations in one subunit can be compensated by stabilizing mutations in the other, keeping binding energy and overall stability near the optimum. While compensatory mutations could also occur at different positions within identical subunits of the HMs (*Uguzzoni et al., 2017*), the HET would have access to those same mutations in addition to combinations of mutations in the two paralogous genes. As a result, the number of available compensatory mutations for the HET would be higher than for the HMs.

Furthermore, FoldX in our simulations predicts a slight overall enrichment towards positive epistasis for mutations affecting the two genes whose effects are combined in the HET. This would also contribute to the retention of the HET without adaptive mutations. Together, the smaller effect sizes

of individual mutations on HET, the expanded number of compensatory mutations, and the mutational bias toward positive epistasis for the HET observed in our simulations suggest that the assembly of HET might be more robust to mutations than that of HMs. Thus, our simulations show higher potential for the specific retention of the HET than for the specific retention of the two HMs. The next step will be to test these models experimentally.

One of our observations is that WGDs present proportionally more HM&HET motifs than SSDs. We propose that this is at least partly due to the age of paralogs, which would lead to more divergence. This proposal was based on the fact that SSDs in yeast show lower sequence conservation and are thus likely older than WGDs and that even among WGDs, homeologs show the HM&HET motif less frequently than HMs compared to true ohnologs, which are by definition younger. However, the mode of duplication itself could also impact HET maintenance. For instance, upon a whole-genome duplication event, all subunits of complexes are duplicated at the same time, which may contribute to the increased retention of WGDs in complexes compared to SSDs and thus maintain HETs. Indeed, small-scale duplications perturb the stoichiometry of complexes whereas whole-genome duplications preserve it (*Birchler and Veitia, 2012*; *Hakes et al., 2007*; *Papp et al., 2003*; *Rice and McLysaght, 2017*). In addition, SSDs display higher evolutionary rates than WGDs (*Fares et al., 2013*), which could lead to the faster loss of their interactions. Another factor that differs is that some WGDs are maintained due to selection for higher gene dosage (*Ascencio et al., 2017*; *Edger and Pires, 2009*; *Gout and Lynch, 2015*; *Sugino and Innan, 2006*; *Thompson et al., 2016*). Therefore, the ancestral gene sequence, regulation and function would be conserved, which ultimately favors the maintenance of HETs among WGDs.

We noticed a significant fraction of paralogs forming only HMs but not HET, including some cases of recent duplicates, indicating that the forces maintaining HETs can be overcome. Moreover, although SSDs are more divergent than WGDs on average, sequence divergence and domain composition differ slightly (not significant) between HMs and HM&HETs, suggesting a mechanism other than amino acid sequence divergence for HET loss. Duplicated genes in yeast and other model systems often diverge quickly in terms of transcriptional regulation (*Li et al., 2005*; *Thompson et al., 2013*) due to *cis* regulatory mutations (*Dong et al., 2011*). Because transcriptional divergence of paralogs can directly change PPI profiles, expression changes would be able to rapidly change a motif from HM&HET to HM. Indeed, switching the coding sequences between paralogous loci is sometimes sufficient to change PPI specificity in living cells (*Gagnon-Arsenault et al., 2013*). Protein localization can also be an important factor affecting the ability of proteins to interact (*Rochette et al., 2014*). We found that paralogs that derive from HMs and that have lost their ability to form HETs are less co-regulated and less co-localized. This divergence suggests that regulatory evolution could play a role in relieving duplicated homomeric proteins from the correlated effects of mutations affecting shared protein interfaces.

Overall, our analyses show that duplication of self-interacting proteins creates paralogs whose evolution is constrained by pleiotropy in ways that are not expected for monomeric paralogs. Pleiotropy has been known to influence the architecture of complex traits and thus to shape their evolution (*Wagner and Zhang, 2011*). However, how it takes place at the molecular level and how it can be overcome to allow molecular traits to evolve independently is still largely unknown. Here, we provide a simple system in which the role of pleiotropy can be examined at the molecular level. Because gene duplication is a major mechanism responsible for the evolution of cellular networks and because a large fraction of proteins are oligomeric, the pleiotropic and epistatic constraints described here could be an important force in shaping protein networks. Another important result is that negative selection for the maintenance of heteromers of paralogs is not needed for their preservation, further enhancing our understanding of the role of non-adaptive evolution in shaping the complexity of cellular structures (*Lynch et al., 2014*).

## Materials and methods

**Key resources table**

*Continued on next page*

Continued

| Reagent type (species) or resource | Designation | Source or reference | Identifiers | Additional information |
|---|---|---|---|---|
| Reagent type (species) or resource | Designation | Source or reference | Identifiers | Additional information |
| Strain, strain background (*Saccharomyces cerevisiae*) | Yeast Protein Interactome Collection - DHFR F[1,2] and DHFR F[3] strains, BY4741 and BY4742 (MATa and MATα) | GE Healthcare Dharmacon Inc, *Tarassov et al., 2008* | Cat. #YSC5849 | See *Supplementary file 2* Tables S9 and S10 for the complete list of strains |
| Strain, strain background (*Saccharomyces cerevisiae*) | DHFR F[1,2] strains, BY4741 (MATa) | *Diss et al., 2017* and this paper | | See *Supplementary file 2* Tables S9 and S10 for the complete list of strains |
| Strain, strain background (*Saccharomyces cerevisiae*) | DHFR F[3] strains, BY4742 (MATα) | *Diss et al., 2017* and this paper | | See *Supplementary file 2* Tables S9 and S10 for the complete list of strains |
| Strain, strain background (*Saccharomyces cerevisiae*) | RY1010, PJ69-4A (MATa) | *Yachie et al., 2016* | | |
| Strain, strain background (*Saccharomyces cerevisiae*) | RY1030, PJ69-4alpha (MATα) | *Yachie et al., 2016* | | |
| Strain, strain background (*Saccharomyces cerevisiae*) | YY3094, PJ69-4A (MATa) | This paper – available from Christian Landry upon request | | |
| Strain, strain background (*Saccharomyces cerevisiae*) | YY3095, PJ69-4alpha (MATα) | This paper – available from Christian Landry upon request | | |
| Strain, strain background (*Lachancea kluyveri*) | *Lachancea kluyveri, CBS 3082* | *Kurtzman, 2003* | | |
| Strain, strain background (*Zygosaccharomyces rouxii*) | *Zygosaccharomyces rouxii, CBS 732* | *Pribylova et al., 2007* | | |
| Strain, strain background (*Escherichia coli*) | MC1061 | CGSC | Cat. #6649 | |
| Recombinant DNA reagent | pAG25-linker-F[1,2]-ADHterm (plasmid) | *Tarassov et al., 2008* | | |
| Recombinant DNA reagent | pAG32-linker-F[3]-ADHterm (plasmid) | *Tarassov et al., 2008* | | |
| Recombinant DNA reagent | pDEST-AD (TRP1) (plasmid) | *Rual et al., 2005* | | |
| Recombinant DNA reagent | pDEST-DB (LEU2) (plasmid) | *Rual et al., 2005* | | |

*Continued on next page*

*Continued*

| Reagent type (species) or resource | Designation | Source or reference | Identifiers | Additional information |
|---|---|---|---|---|
| Recombinant DNA reagent | pDN0501 (TRP1) (plasmid) | This paper – available from Christian Landry upon request | | |
| Recombinant DNA reagent | pDN0502 (LEU2) (plasmid) | This paper – available from Christian Landry upon request | | |
| Recombinant DNA reagent | pHMA1001 (TRP1) (plasmid) | This paper – available from Christian Landry upon request | | |
| Recombinant DNA reagent | pHMA1003 (LEU2) (plasmid) | This paper – available from Christian Landry upon request | | |
| Recombinant DNA reagent | pDEST-DHFR F[1,2] (TRP1) (plasmid) | This paper – available from Christian Landry upon request | | |
| Recombinant DNA reagent | pDEST-DHFR F[1,2] (LEU2) (plasmid) | This paper – available from Christian Landry upon request | | |
| Recombinant DNA reagent | pDEST-DHFR F[3] (TRP1) (plasmid) | This paper – available from Christian Landry upon request | | |
| Recombinant DNA reagent | pDEST-DHFR F[3] (LEU2) (plasmid) | This paper – available from Christian Landry upon request | | |
| Recombinant DNA reagent | pDONR201 (plasmid) | Invitrogen | Cat. #11798–014 | |
| Recombinant DNA reagent | PacI | New England BioLabs Inc | Cat. #R0547S | |
| Recombinant DNA reagent | SacI | New England BioLabs Inc | Cat. #R0156S | |
| Recombinant DNA reagent | SpeI | New England BioLabs Inc | Cat. #R0133S | |
| Recombinant DNA reagent | PI-PspI | New England BioLabs Inc | Cat. #R0695S | |
| Sequence-based reagent | Oligonucleotides | This paper | PCR primers | See *Supplementary file 2* Table S12 for the complete list |
| Sequence-based reagent | DEY011 | Integrated DNA Technologies, Inc | gBlock | See *Supplementary file 2* Table S12 for the sequence |
| Commercial assay or kit | Presto Mini Plasmid Kit | Geneaid Biotech Ltd | Cat. #PDH300 | |
| Commercial assay or kit | Lexogen Quantseq 3' mRNA kit | D-Mark Biosciences | Cat. #012.24A | |
| Commercial assay or kit | Gateway BP Clonase II enzyme mix | Thermo Fisher Scientific | Cat. #11789020 | |

*Continued on next page*

*Continued*

| Reagent type (species) or resource | Designation | Source or reference | Identifiers | Additional information |
|---|---|---|---|---|
| Commercial assay or kit | Gateway LR Clonase II enzyme mix | Thermo Fisher Scientific | Cat. #11791020 | |
| Commercial assay or kit | Gibson Assembly Master Mix | New England BioLabs Inc | Cat. # E2611L | |
| Chemical compound, drug | Kanamycin | BioShop Canada, Inc | Cat. #KAN201.10 | |
| Chemical compound, drug | Ampicillin | BioShop Canada, Inc | Cat. #AMP201 | |
| Chemical compound, drug | Nourseothricin (NAT) | WERNER BioAgents GmbH | Cat. #5.010.000 | |
| Chemical compound, drug | Hygromycin B (HygB) | BioShop Canada, Inc | Cat. #HYG003 | |
| Chemical compound, drug | Methotrexate (MTX) | BioShop Canada, Inc | Cat. #MTX440 | |
| Software, algorithm | MUSCLE v 3.8.31 | *Edgar, 2004* | RRID: SCR_011812 | |
| Software, algorithm | gitter (R package version 1.1.1) | *Wagih and Parts, 2014* | | |
| Software, algorithm | normalmixEM function (R mixtools package) | *Benaglia et al., 2009* | | |
| Software, algorithm | FastQC | *Andrews, 2010* | RRID: SCR_014583 | |
| Software, algorithm | cutadapt | *Martin, 2011* | RRID: SCR_011841 | |
| Software, algorithm | bwa | *Li and Durbin, 2009* | RRID: SCR_010910 | |
| Software, algorithm | HTSeq (Python package) | *Anders et al., 2015* | RRID: SCR_005514 | |
| Software, algorithm | BLASTP (version 2.6.0+) | *Camacho et al., 2009* | RRID: SCR_001010 | |
| Software, algorithm | FoldX suite version 4 | *Guerois et al., 2002* and *Schymkowitz et al., 2005* | RRID: SCR_008522 | |
| Software, algorithm | FreeSASA | *Mitternacht, 2016* | | |
| Software, algorithm | Biopython | *Cock et al., 2009* | RRID: SCR_007173 | |
| Other, database | IntAct | *Orchard et al., 2014* | RRID: SCR_006944 | https://www.ebi.ac.uk/intact/ |
| Other, database | Yeast Gene Order Browser (YGOB) | *Byrne and Wolfe, 2005* | | http://ygob.ucd.ie/ |
| Other, database | PhylomeDB | *Huerta-Cepas et al., 2008* | RRID: SCR_007850 | http://phylomedb.org/ |
| Other, database | Protein Data Bank (PDB) | *Berman et al., 2000* | RRID: SCR_012820 | https://www.rcsb.org/ |

*Continued on next page*

*Continued*

| Reagent type (species) or resource | Designation | Source or reference | Identifiers | Additional information |
|---|---|---|---|---|
| Other, database | Ensembl | *Zerbino et al., 2018* | RRID: SCR_002344 | http://useast.ensembl.org/info/data/ftp/index.html |
| Other, database | TheCellMap (version of March 2016) | *Usaj et al., 2017* | | http://thecellmap.org/ |
| Other, database | Saccharomyces Genome Database (SGD) | *Cherry et al., 2012* | RRID: SCR_004694 | https://www.yeastgenome.org/ |
| Other, database | Complex Portal | *Meldal et al., 2015* | RRID: SCR_015038 | https://www.ebi.ac.uk/complexportal/ |
| Other, database | CYC2008 catalog | *Pu et al., 2009* *Pu et al., 2007* | | http://wodaklab.org/cyc2008/ |
| Other, database | YEASTRACT | *Teixeira et al., 2018*, *Teixeira et al., 2006* | RRID: SCR_006076 | http://www.yeastract.com/ |
| Other, database | Yeast GFP Fusion Localization Database (YeastGFP) | *Huh et al., 2003* | | https://yeastgfp.yeastgenome.org/ |
| Other, database | The Protein Families Database (Pfam) | *El-Gebali et al., 2019* | RRID: SCR_004726 | https://pfam.xfam.org/ |
| Other, database | UniprotKB database | *The UniProt Consortium, 2019* | RRID: SCR_004426 | https://www.uniprot.org/ |
| Other, database | BIOGRID-3.5.166 | *Chatr-Aryamontri et al., 2017*, *Chatr-Aryamontri et al., 2013* | RRID: SCR_007393 | https://thebiogrid.org/ |
| Other, database | Ohnologs | *Singh et al., 2015* | | http://ohnologs.curie.fr/ |
| Other, dataset | Supplementary materials of *Benschop et al. (2010)* | *Benschop et al., 2010* | | https://doi.org/10.1016/j.molcel.2010.06.002 |
| Other, dataset | Supplementary materials of *Kim et al. (2019)* | *Kim et al., 2019* | | https://doi.org/10.1101/gr.231860.117 |
| Other, dataset | Supplementary materials of *Ihmels et al. (2004)* | *Ihmels et al., 2004* | | https://doi.org/10.1093/bioinformatics/bth166 |
| Other, dataset | Supplementary materials of *Gasch et al. (2017)* | *Gasch et al., 2017* | | https://doi.org/10.1371/journal.pbio.2004050 |
| Other, dataset | Supplementary materials of *Guan et al. (2007)* | *Guan et al., 2007* | | https://doi.org/10.1534/genetics.106.064329 |
| Other, dataset | Supplementary materials of *Tarassov et al. (2008)* | *Tarassov et al., 2008* | | https://doi.org/10.1126/science.1153878 |
| Other, dataset | Supplementary materials of *Stynen et al. (2018)* | *Stynen et al., 2018* | | https://doi.org/10.1016/j.cell.2018.09.050 |
| Other, dataset | Supplementary materials of *Lan and Pritchard (2016)* | *Lan and Pritchard, 2016* | | https://doi.org/10.1126/science.aad8411 |

The protein-protein interactions identified in this publication have been submitted to the IMEx (http://www.imexconsortium.org) consortium through IntAct (*Orchard et al., 2014*) and are assigned

the identifier IM-26944. All scripts used to analyze the data are available at https://github.com/land-rylaboratory/Gene_duplication_2019 (*Marchant, 2019*; copy archived at https://github.com/elifes-ciences-publications/Gene_duplication_2019).

## Characterization of paralogs in *S. cerevisiae* genome

### Classification of paralogs by mechanism of duplication

We classified duplicated genes in three categories according to their mechanism of duplication: small-scale duplicates (SSDs); whole-genome duplicates (WGDs) (*Byrne and Wolfe, 2005*); and doubly duplicated (2D, SSDs and WGDs). We removed WGDs from the paralogs defined in *Guan et al. (2007)* to generate the list of SSDs. Among paralog pairs with less than 20% of sequence identity in the multiple sequence alignments (*Edgar, 2004*), we kept only those sharing the same phylome (PhylomeDB; *Huerta-Cepas et al., 2008*) to make sure they were true paralogs. If one of the two paralogs of an SSD pair was associated to another paralog in a WGD pair, this paralog was considered a 2D (*Supplementary file 2* Tables S1 and S2). To decrease the potential bias from multiple duplication events, we removed the 2Ds and paralogs from successive small-scale genome duplications from the data on interaction motifs. We used data from *Marcet-Houben and Gabaldón (2015)* to identify WGDs that are likely true ohnologs or that originated from allopolyploidization (homeologs).

### Sequence similarity

Conversion tables between PhylomeDB IDs and systematic yeast IDs were downloaded from ftp://phylomedb.org/phylomedb/all_id_conversion.txt.gz on May 15th, 2019. Sequence identity was calculated from multiple sequence alignments from phylome 0003 from PhylomeDB (*Huerta-Cepas et al., 2008*). The yeast phylome consists of 60 completely sequenced fungal species, with *Homo sapiens* and *Arabidopsis thaliana* as outgroups. Sequences in these phylomes were aligned with MUSCLE v 3.6. When two paralogs were not found in the same multiple sequence alignment from PhylomeDB (32 pairs out of 462 pairs), the sequences were taken from the reference proteome of *S. cerevisiae* assembly R64-1-1 downloaded on April 16th, 2018 from the Ensembl database at (http://useast.ensembl.org/info/data/ftp/index.html) (*Zerbino et al., 2018*) and realigned to the rest of the phylome with MUSCLE version 3.8.31 (*Edgar, 2004*). For six pairs of paralogs that did not have phylomeDB IDs assigned to them, pairwise alignments of their sequences with MUSCLE version 3.8.31 (*Edgar, 2004*) were used.

### Function, transcription factor binding sites, localization of protein complexes, and Pfam annotations

We obtained GO terms (GO slim) from SGD (*Cherry et al., 2012*) in September 2018. We removed terms corresponding to missing data and created a list of annotations for each SSD and WGD. Annotations were compared to measure the extent of similarity between two members of a pair of duplicates. We calculated the similarity of molecular function, cellular component and biological process taking the number of GO terms in common divided by the total number of unique GO terms of the two paralogs combined (Jaccard index). We compared the same way transcription factor binding sites using YEASTRACT data (*Teixeira et al., 2018*; *Teixeira et al., 2006*), cellular localizations extracted from the YeastGFP database (*Huh et al., 2003*) and many phenotypes associated with the deletion of paralogs (data from SGD in September 2018). For the deletion phenotypes, we kept only information with specific changes (a feature observed and a direction of change relative to wild type). We compared the pairwise correlation of genetic interaction profiles using the genetic interaction profile similarity (measured by Pearson's correlation coefficient) of non-essential genes available in TheCellMap database (version of March 2016) (*Usaj et al., 2017*). We used the median of correlation coefficients if more than one value was available for a given pair. Non-redundant set of protein complexes was derived from the Complex Portal (*Meldal et al., 2015*), the CYC2008 catalog (*Pu et al., 2009*; *Pu et al., 2007*) and (*Benschop et al., 2010*).

We downloaded Pfam domain annotations (*El-Gebali et al., 2019*) for the whole *S. cerevisiae* reference proteome on May 2nd, 2019 from the UniprotKB database (*The UniProt Consortium, 2019*). We removed pairs of paralogs for which at least one of the proteins had no annotated domains and calculated the Jaccard index (*Supplementary file 2* Table S3).

## Homomers and heteromers identified from databases

To complement our experimental data, we extracted HMs and HETs published in BioGRID version BIOGRID-3.5.166 (*Chatr-Aryamontri et al., 2017*; *Chatr-Aryamontri et al., 2013*). We used data derived from the following detection methods: Affinity Capture-MS, Affinity Capture-Western, Reconstituted Complex, Two-hybrid, Biochemical Activity, Co-crystal Structure, Far Western, FRET, Protein-peptide, PCA and Affinity Capture-Luminescence.

It is possible that some HMs or HETs are absent from the database because they have been tested but not detected. This negative information is not reported in databases. We therefore attempted to discriminate non-tested interactions from truly non interacting pairs. A study in which there was not a single HM reported was considered as missing data for all HMs. For both HMs and HETs, the presence of a protein (or both proteins for HET) as both bait and prey but the absence of interaction was considered as evidence for no interaction. Otherwise, it was considered as missing data.

We also considered data from crystal structures. If a HM was detected in the Protein Data Bank (PDB) (*Berman et al., 2000*), we inferred that it was present. If the HM was not detected but the monomer was reported, it is likely that there is no HM for this protein and it was thus considered non-HM. If there was no monomer and no HM, the data were considered as missing. We proceeded the same way for HETs.

Data on genome-wide HM screens was obtained from *Kim et al. (2019)* and *Stynen et al. (2018)*. The two experiments used Protein-fragment complementation assays (PCA), the first one using the dihydrofolate reductase (DHFR) enzyme as a reporter and the second one, a fluorescent protein (also known as Bimolecular fluorescence complementation (BiFC)). We discarded proteins from *Stynen et al. (2018)* flagged as problematic by *Rochette et al. (2014)*; *Stynen et al. (2018)*; *Tarassov et al. (2008)* and false positives identified by *Kim et al. (2019)*. All discarded data was considered as missing data. We examined all proteins tested and considered them as HM if they were reported as positive and as non-HM if tested but not reported as positive.

## Experimental Protein-fragment complementation assay

We performed a screen using PCA based on DHFR (*Tarassov et al., 2008*) following standard procedures (*Rochette et al., 2015*; *Tarassov et al., 2008*). The composition of all following media used in this study is described in *Supplementary file 2* Table S11.

### DHFR strains

We identified 485 pairs of SSDs and 156 pairs of WGDs present in the Yeast Protein Interactome Collection (*Tarassov et al., 2008*) and another set of 155 strains constructed by *Diss et al. (2017)*. We retrieved strains from the collection (*Tarassov et al., 2008*) and we grew them on NAT (DHFR F [1,2] strains) and HygB (DHFR F[3] strains) media. We confirmed the insertion of the DHFR fragments at the correct location by colony PCR using a specific forward Oligo-C targeting a few hundred base pairs upstream of the fusion and a reverse complement oligonucleotide ADHterm_R located in the ADH terminator after the DHFR fragment sequence (*Supplementary file 2* Table S12). Cells from colonies were lysed in 40 µL of 20 mM NaOH for 20 min at 95°C. Tubes were centrifuged for 5 min at 1800 g and 2.5 µL of supernatant was added to a PCR mix composed of 16.85 µL of DNAse free water, 2.5 µL of 10X Taq buffer (BioShop Canada Inc, Canada), 1.5 µL of 25 mM MgCl2, 0.5 µL of 10 mM dNTP (Bio Basic Inc, Canada), 0.15 µL of 5 U/µL Taq DNA polymerase (BioShop Canada Inc, Canada), 0.5 µL of 10 µM Oligo-C and 0.5 µL of 10 µM ADHterm_R. The initial denaturation was performed for 5 min at 95°C and was followed by 35 cycles of 30 s of denaturation at 94°C, 30 s of annealing at 55°C, 1 min of extension at 72°C and by a 3 min final extension at 72°C. We confirmed by PCR 2025 strains from the DHFR collection and 126 strains out of the 154 from *Diss et al. (2017)* (*Supplementary file 2* Tables S9, S10, and S12).

The missing or non-validated strains were constructed *de novo* using the standard DHFR strain construction protocol (*Michnick et al., 2016*; *Rochette et al., 2015*). The DHFR fragments and associated resistance modules were amplified from plasmids pAG25-linker-F[1,2]-ADHterm (NAT resistance marker) and pAG32-linker-F[3]-ADHterm (HygB resistance marker) (*Tarassov et al., 2008*) using oligonucleotides defined in (*Supplementary file 2* Table S12). PCR mix was composed of 16.45 µL of DNAse free water, 1 µL of 10 ng/µL plasmid, 5 µL of 5X Kapa Buffer (Kapa Biosystems,

Inc, A Roche Company, Canada), 0.75 µL of 10 mM dNTPs, 0.3 µL of 1 U/µL Kapa HiFi HotStart DNA polymerase (Kapa Biosystems, Inc, A Roche Company, Canada) and 0.75 µL of both forward and reverse 10 µM oligos. The initial denaturation was performed for 5 min at 95°C and was followed by 32 cycles of 20 s of denaturation at 98°C, 15 s of annealing at 64.4°C, 2.5 min of extension at 72°C and 5 min of a final extension at 72°C.

We performed strain construction in BY4741 (MAT**a** his3Δ leu2Δ met15Δ ura3Δ) and BY4742 (MATα his3Δ leu2Δ lys2Δ ura3Δ) competent cells prepared as in *Gagnon-Arsenault et al. (2013)* for the DHFR F[1,2] and DHFR F[3] fusions, respectively. Competent cells (20 µL) were combined with 8 µL of PCR product (~0.5–1 µg/µL) and 100 µL of Plate Mixture (PEG3350 40%, 100 mM of LiOAc, 10 mM of Tris-Cl pH 7.5 and 1 mM of EDTA). Cells were vortexed and incubated at room temperature without agitation for 30 min. After adding 15 µL of DMSO and mixing thoroughly, heat shock was performed by incubating in a water bath at 42°C for 15–20 min. Following the heat shock, cells were spun down at 400 g for 3 min. Supernatant was removed by aspiration and cell pellets were resuspended in 100 µL of YPD. Cells were allowed to recover from heat shock for 4 hr at 30°C before being plated on NAT (DHFR F[1,2] strains) or HygB (DHFR F[3] strains) plates. Cells were incubated at 30°C for 3 days. The correct integration of DHFR fragments was confirmed by colony PCR as described above and later by sequencing (Plateforme de séquençage et de génotypage des génomes, CRCHUL, Canada) for specific cases where the interaction patterns suggested a construction problem, for instance when the HET was observed in one direction only or when one HM was missing for a given pair. At the end, we reconstructed and validated 146 new strains (*Supplementary file 2* Tables S9 and S10). From all available strains, we selected pairs of paralogs for which we had both proteins tagged with both DHFR fragments (four different strains per pair). This resulted in 1172 strains corresponding to 293 pairs of paralogs (*Supplementary file 2* Tables S9 and S10). We finally discarded pairs considered as leading to false positives by *Tarassov et al. (2008)*, which resulted in 235 pairs.

## Construction of DHFR plasmids for orthologous gene expression

For the plasmid-based PCA, Gateway cloning-compatible destination plasmids pDEST-DHFR F[1,2] (TRP1 and LEU2) and pDEST-DHFR F[3] (TRP1 and LEU2) were constructed based on the CEN/ARS low-copy yeast two-hybrid (Y2H) destination plasmids pDEST-AD (TRP1) and pDEST-DB (LEU2) (*Rual et al., 2005*). A DNA fragment having I-CeuI restriction site was amplified using DEY001 and DEY002 primers (*Supplementary file 2* Table S12) without template and another fragment having PI-PspI/I-SceI restriction site was amplified using DEY003 and DEY004 primers (*Supplementary file 2* Table S12) without template. pDEST-AD and pDEST-DB plasmids were each digested by PacI and SacI and mixed with the I-CeuI fragment (destined to the PacI locus) and PI-PspI/I-SceI fragment (destined to the SacI locus) for Gibson DNA assembly (*Gibson et al., 2009*) to generate pDN0501 (TRP1) and pDN0502 (LEU2). Four DNA fragments were then prepared to construct the pDEST-DHFR F[1,2] vectors: (i) a fragment containing the ADH1 promoter; (ii) a fragment containing a Gateway destination site; (iii) a DHFR F[1,2] fragment; and (iv) a backbone plasmid fragment. The ADH1 promoter fragment was amplified from pDN0501 using DEY005 and DEY006 primers (*Supplementary file 2* Table S12) and the Gateway destination site fragment was amplified from pDN0501 using DEY007 and DEY008 primers (*Supplementary file 2* Table S12). The DHFR-F[1,2] fragment was amplified from pAG25-linker-F[1,2]-ADHterm (*Tarassov et al., 2008*) using DEY009 and DEY010 primers (*Supplementary file 2* Table S12).

The backbone fragment was prepared by restriction digestion of pDN0501 or pDN0502 using I-CeuI and PI-PspI and purified by size-selection. The four fragments were assembled by Gibson DNA assembly where each fragment pair was overlapping with more than 30 bp, producing pHMA1001 (TRP1) or pHMA1003 (LEU2). The PstI–SacI region of the plasmids was finally replaced with a DNA fragment containing an amino acid flexible polypeptide linker (GGGGS) prepared by PstI/SacI double digestion of a synthetic DNA fragment DEY011 to produce pDEST-DHFR F[1,2] (TRP1) and pDEST-DHFR F[1,2] (LEU2). The DHFR F[3] fragment was then amplified from pAG32-linker-F[3]-ADHterm with DEY012 and DEY013 primers (*Supplementary file 2* Table S12), digested by SpeI and PI-PspI, and used to replace the SpeI–PI-PspI region of the pDEST-DHFR F[1,2] plasmids, producing pDEST-DHFR F[3] (TRP1) and pDEST-DHFR F[3] (LEU2) plasmids. In this study, we used pDEST-DHFR F[1,2] (TRP1) and pDEST-DHFR F[3] (LEU2) for the plasmid-based DHFR PCA.

After Gateway LR cloning of Entry Clones to these destination plasmids, the expression plasmids encode protein fused to the DHFR fragments via an NPAFLYKVVGGGSTS linker.

We obtained the orthologous gene sequences for the mitochondrial translocon complex and the transaldolase proteins of *Lachancea kluyveri* (*Kurtzman, 2003*) and *Zygosaccharomyces rouxii* (*Pribylova et al., 2007*) from the Yeast Gene Order Browser (YGOB) (*Byrne and Wolfe, 2005*). Each ORF was amplified from appropriate gDNA using oligonucleotides listed in *Supplementary file 2* Table S12. We used 300 ng of purified PCR product to set a BPII recombination reaction (5 µL) into the Gateway Entry Vector pDONR201 (150 ng) according to the manufacturer's instructions (Invitrogen, USA). BPII reaction mix was incubated overnight at 25˚C. The reaction was inactivated with proteinase K. The whole reaction was used to transform MC1061 competent *E. coli* cells (*Green and Rogers, 2013*), followed by selection on solid 2YT medium supplemented with 50 mg/L of kanamycin (BioShop Inc, Canada) at 37˚C. Positive clones were detected by PCR using an ORF specific oligonucleotide and a general pDONR201 primer (*Supplementary file 2* Table S12). We then extracted the positive Entry Clones using Presto Mini Plasmid Kit (Geneaid Biotech Ltd, Taiwan) for downstream application.

LRII reactions were performed by mixing 150 ng of the Entry Clone and 150 ng of expression plasmids (pDEST-DHFR F[1,2]-TRP1 or pDEST-DHFR F[3]-LEU2) according to manufacturer's instructions (Invitrogen, USA). The reactions were incubated overnight at 25˚C and inactivated with proteinase K. We used the whole reaction to transform MC1061 competent *E. coli* cells, followed by selection on solid 2YT medium supplemented with 100 mg/L ampicillin (BioShop Inc, Canada) at 37˚C. Positive clones were confirmed by PCR using a ORF specific primer and a plasmid universal primer. The sequence-verified expression plasmids bearing the orthologous fusions with DHFR F[1,2] and DHFR F[3] fragments were used to transform the yeast strains YY3094 (MAT**a** *leu2-3,112 trp1-901 his3-200 ura3-52 gal4Δ gal80Δ LYS2::P_{GAL1}-HIS3 MET2::P_{GAL7}-lacZ cyh2^R can1Δ::P_{CMV}-rtTA-KanMX4*) and YY3095 (MATα *leu2-3,112 trp1-901 his3-200 ura3-52 gal4Δ gal80Δ LYS2::P_{GAL1}-HIS3 MET2::P_{GAL7}-lacZ cyh2^R can1Δ::T_{ADH1}-P_{tetO2}-Cre-T_{CYC1}-KanMX4*), respectively. Selection was done on SC -trp -ade (YY3094) or on SC -leu -ade (YY3095). The strains YY3094 and YY3095 were generated from BFG-Y2H toolkit strains RY1010 and RY1030 (*Yachie et al., 2016*), respectively, by restoring their wild type *ADE2* genes. The *ADE2* gene was restored by homologous recombination of the wild type sequence cassette amplified from the laboratory strain BY4741 using primers DEY014 and DEY015 (*Supplementary file 2* Table S12). SC -ade plates were used to obtain successful transformants.

## DHFR PCA experiments

Three DHFR PCA experiments were performed, hereafter referred to as PCA1, PCA2 and PCA3. The configuration of strains on plates and the screenings were performed using robotically manipulated pin tools (BM5-SC1, S&P Robotics Inc, Toronto, Canada; *Rochette et al., 2015*). We first organized haploid strains in 384 colony arrays containing a border of control strains using a cherry-picking 96-pin tool (*Figure 2—figure supplement 7*). We constructed four haploid arrays corresponding to paralog 1 and 2 (P1 and P2) and mating type: MAT**a** P1-DHFR F[1,2]; MAT**a** P2-DHFR F[1,2] (on NAT medium); MATα P1-DHFR F[3]; MATα P2-DHFR F[3] (on HygB medium). Border control strains known to show interaction by PCA (MAT**a** *LSM8*-DHFR F[1-2] and MATα *CDC39*-DHFR F[3]) were incorporated respectively in all MAT**a** DHFR F[1,2] and MATα DHFR F[3] plates in the first and last columns and rows. The strains were organized as described in *Figure 2—figure supplement 7*. The two haploid P1 and P2 384 plates of the same mating type were condensed into a 1536 colony array using a 384-pintool. The two 1536 arrays (one MAT**a** DHFR F[1,2], one MATα DHFR F[3]) were crossed on YPD to systematically test P1-DHFR F[1,2]/P1 DHFR F[3], P1-DHFR F[1,2]/P2-DHFR F[3], P2-DHFR F[1,2]/P1-DHFR F[3] and P2-DHFR F[1,2]/P2-DHFR F[3] interactions in adjacent positions. We performed two rounds of diploid selection (S1 to S2) by replicating the YPD plates onto NAT + HygB and growing for 48 hr. The resulting 1536 diploid plates were replicated twice for 96 hr on DMSO -ade -lys -met control plates (for PCA1 and PCA2) and twice for 96 hr on the selective MTX -ade -lys -met medium (for all runs). Five 1536 PCA plates (PCA1-plate1, PCA1-plate2, PCA2, PCA3-plate1 and PCA3-plate2) were generated this way. We tested the interactions between 277 pairs in five to twenty replicates each (*Supplementary file 2* Table S3).

We also used the robotic platform to generate three bait and three prey 1536 arrays for the DHFR plasmid-based PCA, testing each pairwise interaction at least four times. We mated all MAT**a** DHFR F[1,2] and MATα DHFR F[3] strains on YPD medium at room temperature for 24 hr. We performed two successive steps of diploid selection (SC -leu -trp -ade) followed by two steps on DMSO and MTX media (DMSO -leu -trp -ade and MTX -leu -trp -ade). We incubated the plates of diploid selection at 30°C for 48 hr. Finally, plates from both MTX steps were incubated and monitored for 96 hr at 30°C.

## Analysis of DHFR PCA results

### Image analysis and colony size quantification

All images were analysed the same way, including images from **Stynen et al. (2018)**. Images of plates were taken with a EOS Rebel T5i camera (Canon, Tokyo, Japan) every two hours during the entire course of the PCA experiments. Incubation and imaging were performed in a splmager custom platform (S&P Robotics Inc, Toronto, Canada). We considered images after two days of growth for diploid selection plates and after four days of growth for DMSO and MTX plates. Images were analysed using *gitter* (R package version 1.1.1; **Wagih and Parts, 2014**) to quantify colony sizes by defining a square around the colony center and measuring the foreground pixel intensity minus the background pixel intensity.

### Data filtering

For the images from **Stynen et al. (2018)**, we filtered data based on the diploid selection plates. Colonies smaller than 200 pixels were considered as missing data rather than as non-interacting strains. For PCA1, PCA2 and PCA3, colonies flagged as irregular by *gitter* (as S (colony spill or edge interference) or S, C (low colony circularity) flags) or that did not grow on the last diploid selection step or on DMSO medium (smaller than quantile 25 minus the interquartile range) were considered as missing data. We considered only bait-prey pairs with at least four replicates and used the median of colony sizes as PCA signal. The data was finally filtered based on the completeness of paralogous pairs so we could test HMs and HETs systematically. Thus, we finally obtained results for 241 paralogous pairs (**Supplementary file 2** Tables S3 and S4). Median colony sizes were $\log_2$ transformed after adding a value of 1 to all data to obtain PCA scores. The results of **Stynen et al. (2018)** and PCA1, PCA2 and PCA3 were strongly correlated (**Figure 2—figure supplement 3B**). Similarly, the results correlate well with those reported by **Tarassov et al. (2008)** (**Figure 2—figure supplement 3C**).

### Detection of protein-protein interactions

The distribution of PCA scores was modeled per duplication type (SSD and WGD) and per interaction tested (HM or HET) as in **Diss et al. (2017)** with the *normalmixEM* function (default parameters) available in the R mixtools package (**Benaglia et al., 2009**). The background signal on MTX was used as a null distribution to which interactions were compared. The size of colonies (PCA scores ($PCA_s$)) were converted to z-scores ($Z_s$) using the mean ($\mu_b$) and standard deviation ($sd_b$) of the background distribution ($Z_s = (PCA_s - \mu_b)/sd_b$). PPI were considered detected if $Z_s$ of the bait-prey pair was greater than 2.5 (**Figure 2—figure supplement 8**) (**Chrétien et al., 2018**).

We observed 24 cases in which only one of the two possible HET interactions was detected (P1-DHFR F[1,2] x P2-DHFR F[3] or P2-DHFR F[1,2] x P1-DHFR F[3]). It is typical for PCA assays to detect interactions in only one orientation or the other (See **Tarassov et al. (2008)**). However, this could also be caused by one of the four strains having an abnormal fusion sequence. We verified by PCR and sequenced the fusion sequences to make sure this was not the case. The correct strains were conserved and the other ones were re-constructed and retested. No cases of unidirectional HET were observed in our final results. For all 71 pairs after reconstruction, both reciprocal interactions were detected.

### Dataset integration

The PCA data was integrated with other data obtained from databases. The overlaps among the different datasets and the results of our PCA experiments are shown in **Figure 2—figure supplement 4**.

## Gene expression in MTX condition
### Cell cultures for RNAseq
We used the border control diploid strain from the DHFR PCA experiment (MAT**a**/α *LSM8*-DHFR F [1,2]/*LSM8 CDC39/CDC39*-DHFR F[3]) to measure expression profile in MTX condition. Three over-night pre-cultures were grown separately in 5 ml of NAT + HygB at 30°C with shaking at 250 rpm. A second set of pre-cultures were grown starting from a dilution at $OD_{600}$ = 0.01 in 50 ml in the same condition to an $OD_{600}$ of 0.8 to 1. Final cultures were started at $OD_{600}$ = 0.03 in 250 ml of synthetic media supplemented with MTX or DMSO (MTX -ade -trp -leu or DMSO -ade -trp -leu) at 30°C with shaking at 250 rpm. These cultures were transferred to 5 × 50 ml tubes when they reached an $OD_{600}$ of 0.6 to 0.7 and centrifuged at 1008 g at 4°C for 1 min. The supernatant was discarded and cell pellets were frozen in liquid nitrogen and stored at −80°C until processing. RNA extractions and library generation and amplification were performed as described in *Eberlein et al. (2019)*. Briefly, the Quantseq 3' mRNA kit (Lexogen, Vienna, Austria) was used for library preparation (*Moll et al., 2014*) following the manufacturer's protocol. The PCR cycles number during library amplification was adjusted to 16. The six libraries were pooled and sequenced on a single Ion Torrent chip (Ther-moFisher Scientific, Waltham, United States) for a total of 7,784,644 reads on average per library. Barcodes associated to the samples in this study are listed in *Supplementary file 2* Table S5.

### RNAseq analysis
Read quality statistics were retrieved from the program FastQC (*Andrews, 2010*). Reads were cleaned using cutadapt (*Martin, 2011*). We removed the first 12 bp, trimmed the poly-A tail from the 3' end, trimmed low-quality ends using a cutoff of 15 (phred quality +33) and discarded reads shorter than 30 bp. The number of reads before and after cleaning can be found in *Supplementary file 2* Table S5. Raw sequences can be downloaded under the NCBI BioProject ID PRJNA494421.

Cleaned reads were aligned on the reference genome of S288c from SGD (S288C_reference_ge-nome_R64-2-1_20150113.fsa version) using bwa (*Li and Durbin, 2009*). Because we used a 3'mRNA-Seq Library, reads mapped largely to 3'UTRs. We increased the window of annotated genes in the SGD annotation (saccharomyces_cerevisiae_R64-2-1_20150113.gff version) using the UTR annotation from *Nagalakshmi et al. (2008)*. Based on this reference genes-UTR annotation, the number of mapped reads per genes was estimated using htseq-count of the Python package HTSeq (*Anders et al., 2015*) and reported in *Supplementary file 2* Table S6.

### Correlation of gene expression profiles
The correlation of expression profiles for paralogs was calculated using Spearman's correlation from large-scale microarray data (*Ihmels et al., 2004*) over 1000 mRNA expression profiles from different conditions and different cell cycle phases. These results were compared and confirmed with a large-scale expression data from normalized single-cell RNAseq of *S. cerevisiae* grown in normal or stress-ful conditions (0.7 M NaCl) and from different cell cycle phases (*Gasch et al., 2017*).

## Structural analyses
### Sequence conservation in binding interfaces of yeast complexes
#### Identification of crystal structures
The sequences of paralogs classified as SSDs or WGDs (*Byrne and Wolfe, 2005*; *Guan et al., 2007*) were taken from the reference proteome of *Saccharomyces cerevisiae* assembly R64-1-1 and searched using BLASTP (version 2.6.0+) (*Camacho et al., 2009*) to all the protein sequences con-tained in the Protein Data Bank (PDB) downloaded on September 21st, 2017 (*Berman et al., 2000*). Due to the high sequence identity of some paralogs (up to 95%), their structures were assigned as protein subunits from the PDB that had a match with 100% sequence identity and an E-value lower than 1e-6. Only crystal structures that spanned more than 50% of the full protein length were kept for the following analyses. The same method was used to retrieve PDB structures for human paralo-gous proteins. The human reference proteome Homo_sapiens.GRCh38.pep.all.fa was downloaded on May 16th, 2019 from the Ensembl database (http://useast.ensembl.org/info/data/ftp/index.html) (*Zerbino et al., 2018*). Pairs of paralogs were retrieved from two different datasets (*Lan and*

*Pritchard, 2016*; *Singh et al., 2015*). Protein interactions for those proteins were taken from a merged dataset from the BioGRID (*Chatr-Aryamontri et al., 2017*) and IntAct (*Orchard et al., 2014*) databases. The longest protein isoforms for each gene in the dataset were aligned using BLASTP to the set of sequences from the PDB. Matches with 100% sequence identity and E-values below 1e-6 were assigned to the subunits from the PDB structures.

## Identification of interfaces

Residue positions involved in protein binding interfaces were defined based on the distance of residues to the other subunit (*Tsai et al., 1996*). Contacting residues were defined as those whose two closest non-hydrogen atoms are separated by a distance smaller than the sum of their van der Waals radii plus 0.5 Å. Reference van der Waals radii were obtained with FreeSASA version 2.0.1 (*Mitternacht, 2016*). Nearby residues are those whose alpha carbons are located at a distance smaller than 6 Å. All distances were measured using the Biopython library (version 1.70) (*Cock et al., 2009*).

## Sequence conservation within interfaces

The dataset of PDB files was filtered to include only the crystallographic structures with the highest resolution available for each complex involving direct contacts between subunits of paralogs. Full-length protein sequences from the reference proteome were then aligned to their matching subunits from the PDB with MUSCLE version 3.8.31 (*Edgar, 2004*) to assign the structural data to the residues in the full-length protein sequence. These full-length sequences were then aligned to their paralogs and sequences from PhylomeDB (phylome 0003) (*Huerta-Cepas et al., 2008*) with MUSCLE version 3.8.31. Only three pairs of paralogs that needed realignment were included in this analysis. Sequence identity was calculated within interface regions, which considered the contacting and nearby residues. Paralogs were classified as HM or HM&HET based on the data shown in *Supplementary file 2* Table S3. PDB identifiers for structures included in this analysis are shown in *Supplementary file 2* Table S13. Pairs of paralogs for which the crystallized domain was only present in one of the proteins were not considered for this analysis.

A similar procedure was applied to the human proteins, with sequences aligned to their corresponding PhylomeDB phylogenies from phylome 0076 resulting from forward and reverse alignments obtained with MUSCLE 3.8, MAFFT v6.712b and DIALIGN-TX, and merged with M-COFFEE (*Huerta-Cepas et al., 2008*). Considering that human genes code for multiple isoforms, we took the isoforms from the two paralogs that had the highest sequence identity with respect to the PDB structure. When a gene coded for multiple isoforms that were annotated with identical protein sequence in the human reference proteome, we only kept one of them. This resulted in a set of 40 HM interfaces and 25 HM&HET interfaces for a total of 54 different pairs (35 HM pairs and 19 HM&HET). Pairs of paralogs were classified as HM or HM&HET based on the data in *Supplementary file 2* Tables S14 and S15.

## Simulations of coevolution of protein complexes

### Mutation sampling during evolution of protein binding interfaces

Simulations were carried out with high-quality crystal structures of homodimeric proteins from PDB (*Berman et al., 2000*). Four of them (PDB: 1M38, 2JKY, 3D8X, 4FGW) were taken from the above data set of structures that matched yeast paralogs and two others from the same tier of high-quality structures (PDB: 1A82, 2O1V). The simulations model the duplication of the gene encoding the homodimer, giving rise to separate copies that can accumulate different mutations, leading to the formation of HMs and HETs as in *Figure 1*.

Mutations were introduced using a transition matrix whose substitution probabilities consider the genetic code and allow only substitutions that would require a single base change in the underlying codons (*Thorvaldsen, 2016*). Due to the degenerate nature of the genetic code, the model also allows synonymous mutations. Thus, the model explores the effects of amino acid substitutions in both loci, as well as in one locus only. The framework assumes equal mutation rates at both loci, as it proposes a mutation at each locus after every step in the simulation, with 50 replicate populations of 200 steps of substitution in each simulation. Restricting the mutations to the interface maintains sequence identity above 40%, which has been described previously as the threshold at which protein fold remains similar (*Addou et al., 2009*; *Todd et al., 2001*; *Wilson et al., 2000*).

## Implementation of selection

Simulations were carried out using the FoldX suite version 4 (*Guerois et al., 2002*; *Schymkowitz et al., 2005*). Starting structures were repaired with the RepairPDB function, mutations were simulated with BuildModel followed by the Optimize function, and estimations of protein stability and binding energy of the complex were done with the Stability and Analyse Complex functions, respectively. Effects of mutations on complex fitness were calculated using methods previously described (*Kachroo et al., 2015*). The fitness of a complex was calculated from three components based on the stability of protein subunits and the binding energy of the complex using *Equation 1*:

$$x_i^k = -\log\left[e^{\beta\left(\Delta G_i^k - \Delta G_{threshold}^k\right)} + 1\right]$$ (1)

where $i$ is the index of the current substitution, $k$ is the index of one of the model's three energetic parameters (stability of subunit A, stability of subunit B, or binding energy of the complex), $x_i^k$ is the fitness component of the $k^{th}$ parameter for the $i^{th}$ substitution, $\beta$ is a parameter that determines the smoothness of the fitness curve, $\Delta G_i^k$ is the free energy value of the $k^{th}$ free energy parameter (stability of subunit A, stability of subunit B, or binding energy of the complex) for the $i^{th}$ substitution, and $\Delta G_{threshold}^k$ is a threshold around which the fitness component starts to decrease. The total fitness of the complex after the $i^{th}$ mutation was calculated as the sum of the three computed values for $x_i^k$, as shown in *Equation 2*:

$$x_i = \sum_{k=1}^{3} x_i^k$$ (2)

The fitness values of complexes were then used to calculate the probability of fixation (pfix) or rejection of the substitutions using the Metropolis criterion, as in *Equation 3*:

$$p_{fix} = \begin{cases} 1, if\, x_j > x_i \\ e^{-2N\left(x_i - x_j\right)}, if\, x_j \leq x_i \end{cases}$$ (3)

where $p_{fix}$ is the probability of fixation, $x_i$ is the total fitness value for the complex after $i$ substitutions; $x_j$ is the total fitness value for the complex after $j$ substitutions, with $j = i + 1$; and $N$ is the population size, which influences the efficiency of selection.

Different selection scenarios were examined depending on the complexes whose binding energy and subunit stabilities were under selection: neutral evolution (no selection applied on subunit stability and on the binding energy of the complex), selection on one homodimer, selection on the two homodimers, and selection on the heterodimer. $\beta$ was set to 10, $N$ was set to 1000 and the $\Delta G_{threshold}^k$ were set to 99.9% of the starting values for each complex, following the parameters described in *Kachroo et al. (2015)*. For the simulations with neutral evolution, $\beta$ was set to 0. For simulations with other combinations of parameters, we varied $\beta$ and $N$, one at a time, with $\beta$ taking values of 1 and 20 and $N$ taking values of 100 and 10000. The simulations with 500 substitutions were carried out with $\beta$ set to 10, and $N$ set to 1000.

## Analyses of simulations

The results from the simulations were then analyzed by distinguishing mutational steps with only one non-synonymous mutation (single mutants, between 29% and 34% of the steps in the simulations) from steps with two non-synonymous mutations (double mutants, between 61% and 68% of the steps). The global data was used to follow the evolution of binding energies of the complexes over time, which are shown in *Figure 4*. The effects of mutations in HM and HET were compared using the single mutants (*Figure 5—figure supplement 1*). The double mutants were used to analyze epistatic and pleiotropic effects (*Figure 5*, *Figure 5—figure supplement 3*) and to compare the rates of mutation fixation based on their effects on the HMs (*Figure 5—figure supplement 2*).

## Acknowledgements

This work was supported by Canadian Institutes of Health Research grants 299432, 324265 and 387697 to CRL. AM was supported by a FRQS postdoctoral scholarship. AFC was supported by

fellowships from PROTEO, MITACS, and Université Laval, as well as joint funding from MEES and AMEXCID. SA was supported by an NSERC undergraduate scholarship. CRL holds the Canada Research Chair in Evolutionary Cells and Systems Biology. We thank SW Michnick for sharing data before publication. The authors thank Philippe Després, Johan Hallin and Anna Fijarczyk for comments on the paper, Rohan Dandage for both comments on the paper and assistance on gathering the data for human paralogs, Rong Shi for useful discussions, and Stéphane Larose for assistance on data management.

## Additional information

### Competing interests

Christian R Landry: Reviewing editor, *eLife*. The other authors declare that no competing interests exist.

### Funding

| Funder | Author |
|---|---|
| Fonds de Recherche du Québec - Santé | Axelle Marchant |
| Natural Sciences and Engineering Research Council of Canada | Simon Aubé |
| Canadian Institutes of Health Research | Axelle Marchant<br>Angel F Cisneros<br>Alexandre K Dubé<br>Isabelle Gagnon-Arsenault<br>Diana Ascencio<br>Honey Jain<br>Simon Aubé<br>Chris Eberlein<br>Christian R Landry |

The funders had no role in study design, data collection and interpretation, or the decision to submit the work for publication.

### Author contributions

Axelle Marchant, Conceptualization, Data curation, Formal analysis, Validation, Investigation, Visualization, Writing—original draft, Writing—review and editing; Angel F Cisneros, Data curation, Software, Formal analysis, Validation, Investigation, Visualization, Writing—original draft, Writing—review and editing; Alexandre K Dubé, Conceptualization, Resources, Supervision, Validation, Project administration, Writing—review and editing; Isabelle Gagnon-Arsenault, Conceptualization, Resources, Data curation, Supervision, Validation, Investigation, Project administration, Writing—review and editing; Diana Ascencio, Formal analysis, Validation, Investigation, Visualization, Writing—review and editing; Honey Jain, Formal analysis; Simon Aubé, Chris Eberlein, Methodology; Daniel Evans-Yamamoto, Resources, Methodology; Nozomu Yachie, Resources; Christian R Landry, Conceptualization, Data curation, Formal analysis, Supervision, Funding acquisition, Writing—original draft, Project administration, Writing—review and editing

### Author ORCIDs

Axelle Marchant (iD) https://orcid.org/0000-0001-7134-9769
Angel F Cisneros (iD) https://orcid.org/0000-0002-2030-3653
Alexandre K Dubé (iD) https://orcid.org/0000-0001-8718-9894
Isabelle Gagnon-Arsenault (iD) https://orcid.org/0000-0003-2661-1929
Diana Ascencio (iD) https://orcid.org/0000-0003-4808-1244
Honey Jain (iD) https://orcid.org/0000-0002-1737-9833
Simon Aubé (iD) https://orcid.org/0000-0002-7078-6227
Chris Eberlein (iD) https://orcid.org/0000-0002-8269-5525

Daniel Evans-Yamamoto (iD) http://orcid.org/0000-0001-6467-3827
Nozomu Yachie (iD) https://orcid.org/0000-0003-1582-6027
Christian R Landry (iD) https://orcid.org/0000-0003-3028-6866

## Decision letter and Author response
Decision letter https://doi.org/10.7554/eLife.46754.037
Author response https://doi.org/10.7554/eLife.46754.038

## Additional files

### Supplementary files
• Supplementary file 1. Supplementary text on the performance of PCA as compared to other methods and descriptions of the supplementary tables.
DOI: https://doi.org/10.7554/eLife.46754.031

• Supplementary file 2. Supplementary tables for this work. Table descriptions can be found in *Supplementary file 1*.
DOI: https://doi.org/10.7554/eLife.46754.032

• Transparent reporting form
DOI: https://doi.org/10.7554/eLife.46754.033

### Data availability
All data and scripts are available in the supplementary material or through links that are provided.

The following dataset was generated:

| Author(s) | Year | Dataset title | Dataset URL | Database and Identifier |
|---|---|---|---|---|
| Marchant A | 2018 | RNAseq | https://www.ncbi.nlm.nih.gov/bioproject/494421 | NCBI BioProject, PRJNA494421 |

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
