## [Decision Letter]

Thank you for submitting your article "The role of structural pleiotropy and regulatory evolution in the retention of heteromers of paralogs" for consideration by *eLife*. Your article has been reviewed by Patricia Wittkopp as the Senior Editor and Reviewing Editor, and two reviewers. The following individual involved in review of your submission has agreed to reveal his identity: Jonathan Wells (Reviewer #2).

The reviewers have discussed the reviews with one another and the Reviewing Editor has drafted this decision to help you prepare a revised submission.

Summary:

This is a nice study that tackles an interesting question: namely, does the quaternary structure of an ancestral protein constrain the evolution of subsequent paralogs? The central hypothesis of the paper is that, in cases where the ancestral protein is homomeric, selection to maintain binding interfaces between newly duplicated paralogs will lead to a decrease in the rate of functional divergence of those genes.

In testing this hypothesis, the authors arrive at three key findings: Firstly, heteromeric paralogs of homomeric proteins are common, and are functionally more similar than paralogues of monomeric proteins. Secondly, in silico evolution suggests that negative selection acting on homomeric interfaces is sufficient to maintain heteromeric interactions between paralogs, but if selection acts only on one paralog, then the heteromeric interaction will slowly be lost. Finally, they show that diverging regulatory evolution (e.g. cell localization) can lead to relaxation of the structural constraints, thus enabling functional divergence.

Essential revisions:

1) Modeling of selection. The authors use the method previously described in Kachroo et al., 2015 to calculate the probability of fixation of new mutations; this is "an efficient implementation" of a model described by Sella and Hirsch, PNAS (2005). According to Kachroo et al., equation 3 is accurate as long as the product of the mutation rate and effective population size, N, is very small. Whilst this assumption is generally valid for wild yeast populations, in this study N is set to 1000 – several orders of magnitude lower than is realistic (Tsai et al., 2008). Using more plausible values for N, equation 3 would essentially guarantee fixation for beneficial mutations and vice versa, over-simplifying things. To address this issue, the authors should justify their choice of model and associated parameters and, ideally, demonstrate that their results are robust to changes in these parameters. It would be interesting to see if this affects the "selection on HET AB" case.

2) Analyses of age of duplication. Please clarify how the age of WGD paralogs was calculated, and whether this differs to the method used to calculate SSD ages. Are the two directly comparable? If not, then it might affect some of the conclusions (e.g. subsection “Paralogous heteromers frequently derive from ancestral homomers”). Similarly, people might take issue with the assumption that evolutionary rates are the same for SSDs vs. WGDs. A useful paper here might be Zhu et al., 2013.

3) Analysis of sequence divergence. I include one reviewer's description of this concern in its entirety, but both reviewers agreed with this concern: "In Figure 2E, fewer SSDs form HETs in general, compared to WGDs. This is probably related to the age of duplication events, as the authors note. The two groups of WGDs have the same age. But the SSDs would be from many different times. The authors mention that most SSDs are older, but it seems that some should still be relatively very young. Assuming that an ancestral gene whose protein homodimerizes undergoes a duplication event, the two duplicates should both homodimerize and heterodimerize among them. Accordingly, very young duplicates should belong mostly to group HM&HET. As time goes by, mutations and selection may separate them in two proteins that form only homodimers (group HM), or one of them still homodimerizes and the other evolves towards heterodimerization-only with the other paralogue. In Figure 2F, these two different cases of HM&HET are merged in one group.

I have major concerns about the sequence divergence analyses and their conclusions. First of all, we know that intrinsically disordered regions evolve fast, compared to well conserved domains. Also, some regions may function as flexible linkers (that also evolve fast) between domains. One protein family may evolve fast and another protein family may be very well conserved, irrespective of protein interactions. How do the authors control for this fact?

Moreover, the pleiotropic effect should be on the interaction surface or more broadly on the interaction domain that is responsible for the formation of the homomer or HTs. Usually, this is a well-defined domain or two. Usually, this interaction domain is one of the well conserved regions of the protein and many times a small part of it. I can't imagine how a sequence divergence analysis of the whole protein is meaningful. Maybe a PFAM analysis of the pairs of paralogues and inclusion only of the interacting domains instead of the whole protein? This is a problem. The crystallographic structures analysis they did in subsection “Paralogous heteromers frequently derive from ancestral homomers” is trying to address this problem, but I feel it may not be enough. Another concern is that intrinsically disordered regions are usually involved in transient interactions whereas domains are usually involved in more stable interactions, although this is not an absolute rule. Is this accounted for by the authors in their sequence divergence analyses? Probably not.

In my view, the level of sequence divergence of two paralogues is affected by their time of divergence, but also it is affected by the domain architecture of the protein and whether the interaction is transient or stable. Thus, the authors may need to control for them as well. Basically, the interaction surface/domain is under certain constraints. But other parts of the protein may evolve fast or slow for many other reasons.

Similar concerns exist for functional similarity analyses with GO, phenotypes and genetic interactions. A protein may have more than one functions that may be irrelevant with the formation of HMs and HETs. High functional similarity could only be due to short time after divergence. How do the authors control for that? In my opinion, although some statistically significant differences exist in the analyses of Figure 3, the final message is not clear and strong. "

4) In Figure 4, the positive and negative controls (panel B and C respectively) both behave as expected. However, I was surprised that selection to maintain the heteromer (D) appeared to be a stable state, as there seems to be no obvious reason why the homomers could not eventually be lost. In figures 4 and S9 it seems that the "selection on HET AB" panels seem to be noisier – is this coincidental?

5) This paper integrates data from many sources, which is a strong point. But at the same time, this makes it a lengthy paper, perhaps with too many analyses. At some points, it is easy to lose the main message of the paper and why the authors were doing a particular analysis. Please make the paper more clear and concise, possibly putting details of some analyses (or even some entire analyses) in the supplementary material.

6) Second paragraph of Results section discusses the effect of expression on detecting HMs by PCA. Since expression has an effect on detection of PPIs, is the difference of HMs among singletons, SSDs and WGDs (mentioned in subsection “Homomers and heteromers in the yeast PPI network”) due to this reason? Would it be feasible for the authors to collect subsets of singletons, SSDs and WGDs with similar magnitudes of expression (use bins) and check difference of HMs for these 3 controlled subsets?

7) Please clarify statistics in Supplementary file 2—table S5. More information should be included in that worksheet or somewhere else.

8) In Figure 2F, although there are some statistically significant differences, the various groups span similar orders of magnitude. Please comment on this observation.

9) Optional: Is it feasible for the authors to do an extra series of wet-lab experiments and experimentally test the HMs and HETs of a selected protein with crystal structure that underwent simulated evolution with the different selection scenarios? That would strengthen the paper further.

---

## [Author Response]

We thank Dr Wittkopp and the two reviewers for their helpful comments on our manuscript. We have carefully looked at the comments and used them to improve the quality of our work. Answers to each of their points and further analyses are provided below.

Essential revisions:1) Modeling of selection. The authors use the method previously described in Kachroo et al., 2015 to calculate the probability of fixation of new mutations; this is "an efficient implementation" of a model described by Sella and Hirsch, PNAS (2005). According to Kachroo et al., equation 3 is accurate as long as the product of the mutation rate and effective population size, N, is very small. Whilst this assumption is generally valid for wild yeast populations, in this study N is set to 1000 – several orders of magnitude lower than is realistic (Tsai et al., 2008). Using more plausible values for N, equation 3 would essentially guarantee fixation for beneficial mutations and vice versa, over-simplifying things. To address this issue, the authors should justify their choice of model and associated parameters and, ideally, demonstrate that their results are robust to changes in these parameters. It would be interesting to see if this affects the "selection on HET AB" case.

As stated by the reviewers, this model works under the weak mutation – strong selection assumption that allows every new mutation to fix or disappear before the next one appears. An alternative is to allow populations to accumulate polymorphism but in this case the simplifications that allow to estimate the probability of fixation of a mutation do not longer work. This is why many models work under these assumptions.

The parameters we used (β = 10, N = 1000) capture this effect well since fitness decays rapidly with increments of the deltaG of folding and the deltaG of complex formation, which would be expected to have negative impacts on protein function. However, we agree that N=1000 could appear rather small for yeast populations, although local inbreeding could be high in yeast and allow for the fixation of slightly deleterious mutations (Doniger et al., 2008). Therefore, as suggested by the reviewer, we tested our modeling approach with different combinations of the β and N parameters to verify if our findings were robust to these changes. We observed a very similar behavior with different combinations of parameters. In the light of this, our findings are robust to changes in the model’s parameters, which we expect to generalize over increasing efficiencies of selection up to the population sizes proposed in the paper mentioned by the reviewer. We also ran simulations for 500 substitutions attempted and observed the same overall trends about the slow destabilization of the HMs in the “selection on HET AB” scenario. These results are now presented in Figure 4—figure supplement 3.

2) Analyses of age of duplication. Please clarify how the age of WGD paralogs was calculated, and whether this differs to the method used to calculate SSD ages. Are the two directly comparable? If not, then it might affect some of the conclusions (e.g. subsection “Paralogous heteromers frequently derive from ancestral homomers”). Similarly, people might take issue with the assumption that evolutionary rates are the same for SSDs vs. WGDs. A useful paper here might be Zhu et al., 2013.

We agree with this statement. We chose not to use the Zhu et al., 2013 data because they did not actually calculate the time of divergence. Since they used WGD pairs, they assumed that time is constant for all pairs, which is an assumption we make only partially because we assume that some WGD pairs are in fact homeologs. The method used in the first submission to calculate the age of paralogs was based on the position of the proteins in the phylogeny and suggested that many SSDs are older than the WGDs. Because of the uncertainty in dating the age of paralogs and because it is not critical for our paper, we now removed the emphasis on the age estimation and only use sequence identity as a rough proxy for the age of paralogs, which gives about the same signal. As expected, we show a higher proportion of low sequence identity pairs for SSDs compared to WGDs. This can be seen in the new Figure 2—figure supplement 5A. We further filtered the set of paralogs to make sure the low identify pairs were actual paralogs. We therefore only considered SSDs with sequence identify below 20% if they were in the same phylome in PhylomeDB.

We also added sentences talking about difference of selection pressure on sequence divergence between SSDs and WGDs in the Results section:

“We hypothesize that since SSDs have appeared at different evolutionary times, many of them could be older than WGDs, which could be accompanied by a loss of interactions between paralogs.”

and in the Discussion section:

“In addition, Fares et al. (Fares et al., 2013) suggested that SSDs display higher evolutionary rates than WGDs, which could lead to the loss of their interactions.”

3) Analysis of sequence divergence. I include one reviewer's description of this concern in its entirety, but both reviewers agreed with this concern: "In Figure 2E, fewer SSDs form HETs in general, compared to WGDs. This is probably related to the age of duplication events, as the authors note. The two groups of WGDs have the same age. But the SSDs would be from many different times. The authors mention that most SSDs are older, but it seems that some should still be relatively very young. Assuming that an ancestral gene whose protein homodimerizes undergoes a duplication event, the two duplicates should both homodimerize and heterodimerize among them. Accordingly, very young duplicates should belong mostly to group HM and HET. As time goes by, mutations and selection may separate them in two proteins that form only homodimers (group HM), or one of them still homodimerizes and the other evolves towards heterodimerization-only with the other paralogue. In Figure 2F, these two different cases of HM and HET are merged in one group.

We agree that time of divergence could influence whether pairs from the HM&HET group retain one (1HM&HET) or two HMs (2HM&HET). We originally decided to merge the different cases (1HM and 2HM; 1HM&HET and 2HM&HET) since that would allow for comparing the effect of sequence identity specifically on the formation of the HET. We agree that it could be interesting to detail HM&HET motifs by separating one or two HMs. However, using age group defined in our first submission, only 21 SSD pairs were associated with age group 1 (the younger, similar to WGD). Among them, we observed a majority of pairs without interaction (so not from an ancestral HM) and only two pairs showed the HM&HET interaction motif, so, unfortunately, not enough data is available to perform a comparison. Similarly, only 3 SSDs of HM&HET motif have a high sequence similarity (>69.5% ) in our data, suggesting that we do not have very young paralogs in our dataset. Finally, we would like to confirm that the intuition of the reviewer is right. We do see that at least in cases where we compare 1HM and 2HM, there is more sequence conservation for the 2HM ones for SSDs and WGDs, and more conservation for 2HM&HET than 1HM&HET for WGDs, suggesting that younger paralogs are more likely to show 2 HMs than only 1 HM. This can be seen in Figure 2—figure supplement 5C. However, because we do not have enough data and the signal is rather weak and qualitative, we did not include further discussions on this issue.

I have major concerns about the sequence divergence analyses and their conclusions. First of all, we know that intrinsically disordered regions evolve fast, compared to well conserved domains. Also, some regions may function as flexible linkers (that also evolve fast) between domains. One protein family may evolve fast and another protein family may be very well conserved, irrespective of protein interactions. How do the authors control for this fact?Moreover, the pleiotropic effect should be on the interaction surface or more broadly on the interaction domain that is responsible for the formation of the homomer or HTs. Usually, this is a well-defined domain or two. Usually, this interaction domain is one of the well conserved regions of the protein and many times a small part of it. I can't imagine how a sequence divergence analysis of the whole protein is meaningful. Maybe a PFAM analysis of the pairs of paralogues and inclusion only of the interacting domains instead of the whole protein? This is a problem. The crystallographic structures analysis they did in subsection “Paralogous heteromers frequently derive from ancestral homomers” is trying to address this problem, but I feel it may not be enough. Another concern is that intrinsically disordered regions are usually involved in transient interactions whereas domains are usually involved in more stable interactions, although this is not an absolute rule. Is this accounted for by the authors in their sequence divergence analyses? Probably not.

We agree that different regions within proteins evolve at different rates depending on their function and structure and that, when available, the interacting domains should be looked at (which we did, see below). Nevertheless, looking at the full protein sequences can still provide helpful information and reflect the overall divergence. Indeed, the Pearson correlation coefficient between the yeast paralogs’ sequence identity over the full sequence and within interfaces only is very strong (r = 0.94) for the structures we analyzed. Since the structures of many protein complexes are not available in the PDB because they have not yet been solved, it is hard to determine the specific domains that mediate the interactions. Besides, as disordered residues compose less than 30% of the sequence of most proteins (van der Lee et al., 2014), divergence within domains would account for most of the variation observed in the full sequence analysis.

Nevertheless, we believe it is important to show the sequence identity analyses both over the full sequence and within interfaces. Thus, we decided to move the analyses on crystallographic structures for yeast paralogs from Figure S6 in the previous version to the main paper (Figure 2G and 2H) to highlight how the trends observed for the full sequences reflect those observed for the interacting domains. Since the best way to study interacting domains is based on the crystal structures, we also extended this analysis to pairs of human paralogs from different datasets (Singh et al., 2015; Lan et al., 2016). Higher sequence identity at the interface than for the full sequence was also observed for this dataset, which highlights the evolutionary constraints on interfaces mentioned by the reviewer. However, we did not observe the increase in the ratio of conservation of interfaces to non-interfaces within the crystallized part of the structure. The most likely explanation for the disappearance of this signal is the greater potential for regulatory evolution in humans with respect to yeast. In addition to separating proteins in different subcellular locations, human paralogs can be expressed in specific tissues, as shown by their involvement in tissue-specific diseases (Barshir et al., 2018). These results are shown in Figure 2—figure supplement 6. As a final test, we calculated the Pearson correlation coefficient between the human paralogs’ sequence identity over the full sequence and only within interfaces, and it was also very strong (r = 0.71). As a consequence, we believe that using pairwise sequence identity of the whole protein reflects to a great extent the relative conservation of the interfaces as well. In the future, we would like to extend our analyses on specific protein regions that may be causing a change from one motif to another, but we believe this is beyond the scope of this paper.

Similarly, as suggested by the reviewer, we analyzed the similarity of Pfam domain annotations of the pairs of yeast paralogs. We found that most of the pairs of paralogs (367 out of 448) share all their domains. There is a slight trend for interacting paralogs to share more of their domain annotations than the ones that do not (Figure 3—figure supplement 1A). These results suggest that it is possible that interactions are lost because of the degeneration of interacting domains or the loss of these domains. However, this effect is captured by the full sequence identity of the pairs of paralogs (Figure 3—figure supplement 1B). We added the following paragraph to the main text (subsection “Paralogous heteromers frequently derive from ancestral homomers”):

“Considering that stable interactions are often mediated by protein domains, we looked at the domain composition of paralogs using the Protein Families Database (Pfam) (El-Gebali et al., 2019) We tested if differences in domain composition could explain the frequency of different interaction motifs. We found that 367 of 448 pairs of paralogs (82%) shared all their domain annotations (Table S3). Additionally, HM&HET paralogs tend to have more domains in common but the differences are non-significant and appear to be caused by overall sequence divergence (Figure 3—figure supplement 1A-B). Domain gains and losses are therefore unlikely to contribute to the loss of HET complexes following the duplication of homomers. “

In my view, the level of sequence divergence of two paralogues is affected by their time of divergence, but also it is affected by the domain architecture of the protein and whether the interaction is transient or stable. Thus, the authors may need to control for them as well. Basically, the interaction surface/domain is under certain constraints. But other parts of the protein may evolve fast or slow for many other reasons.

With respect to the type of interactions, the direct interactions registered in the BioGRID, IntAct, and PDB databases employ methods (PCA, crystallography, etc.) best suited for stable interactions. In particular, PCA is specific to stable interactions since the reconstitution of the DHFR enzyme is necessary for the yeast colonies to grow in medium with methotrexate. If the proteins interacted transiently, the transient reconstitution of the DHFR enzyme would not support their growth. As such, we would not expect our results to be affected by transient interactions.

As described previously in Answers 3.1 and 3.2, the structures for many protein complexes have not been solved. This leads to small sample sizes for our analyses of interface conservation. However, since this is the most reliable way to know what the interaction domains are, we believe it is the best way to evaluate sequence divergence of the interfaces. We would also like to emphasize that we use sequence information only to approximate age of paralogs, which is roughly speaking the only way to estimate their age apart from resolving phylogenetic trees (also based on sequences), and not to identify the causal mutations for the loss of HMs and HETs. We agree that this is a very interesting question and we intend on pursuing it in the future.

Similar concerns exist for functional similarity analyses with GO, phenotypes and genetic interactions. A protein may have more than one functions that may be irrelevant with the formation of HMs and HETs. High functional similarity could only be due to short time after divergence. How do the authors control for that? In my opinion, although some statistically significant differences exist in the analyses of Figure 3, the final message is not clear and strong. "

We agree that sequence divergence should be taken into account to compare functional similarity between HM and HM&HET. We included GLM tests with results reported in Supplementary file 2-table S7 (Table S5 in the original submission). We now include graphical representations in Figure 3—figure supplement 4 and Figure 6—figure supplement 4. These show that the extent of similarity is not entirely explained by sequence similarity. In addition, we agree that there are many factors that contribute to functional similarity between paralogous proteins, the fact that they preserved the ability to assemble physically is only one of them. We believe this may not be the strongest factor affecting functional divergence but we do see that it does so in a significant manner and we propose a mechanism for it based on simple principles. We are confident that this is a novel consideration in the study of gene duplication and that it will impact how models and analyses of functional divergence will be constructed in the future.

Functional similarity (GO, phenotype, correlation of genetic interaction, localization and transcription factor) as a function of pairwise amino acid sequence identity for HM motifs (pink) and HM&HET motifs (purple).

4) In Figure 4, the positive and negative controls (panel B and C respectively) both behave as expected. However, I was surprised that selection to maintain the heteromer (D) appeared to be a stable state, as there seems to be no obvious reason why the homomers could not eventually be lost. In Figure 4 and Figure S9 it seems that the "selection on HET AB" panels seem to be noisier – is this coincidental?

The noise in the panels stems from the complexes that are not subject to selection. As shown in Figure 4—figure supplement 4 and Figure 5—figure supplement 2 the effects of mutations on the HMs tend to have a greater magnitude than the effects of mutations on HETs. Therefore, HMs, when not under selection, would be subject to greater variability and would be noisier than the HET when it is not under selection. This agrees with the observations from some of our references (Lukatsky et al., 2006; Lukatsky et al., 2007; André et al., 2008). Furthermore, we observed a slight enrichment in positive epistasis for the HET (weaker effects than expected based on the effects on the HMs), which would also contribute to the HMs being noisier than the HET (see answer minor 1). We now talk about this in the Results section:

“Additionally, mutations tend to have greater effects on the HM than on the HET, which agrees with observations on HMs having a greater variance of binding energies than HETs (Lukatsky et al. 2007; Lukatsky et al. 2006; André et al. 2008). As a consequence, HMs that are not under selection in our simulations show higher variability in their binding energy than HETs that are not under selection.”

Regarding the eventual loss of the HMs under selection for HET, this is expected to be a slow process. The two HMs are very slightly destabilized in this scenario with the numbers of tested mutations because destabilizing mutations are also destabilizing for the HET that is under negative selection. A longer time may achieve heteromeric specificity, but the pleiotropic effects of mutations causes it to advance more slowly than the scenarios with selection on one HM, as shown in Figure 4—figure supplement 3 and discussed in answer 1.

5) This paper integrates data from many sources, which is a strong point. But at the same time, this makes it a lengthy paper, perhaps with too many analyses. At some points, it is easy to lose the main message of the paper and why the authors were doing a particular analysis. Please make the paper more clear and concise, possibly putting details of some analyses (or even some entire analyses) in the supplementary material.

We agree that the main message should not be lost among too many analyses and data sources. We made several modifications:

1) We choose to simplify all figures by focusing only on pairs from the duplication of a potential ancestral HM: HM&HET versus HM without other motifs.

2) We focus on the sequence divergence instead of defining the age of duplication.

3) We moved two paragraphs about the result of comparison of our PCA results with previous interaction studies to the Supplementary material (subsection “Comparison of PCA results with previous studies”):

“The yeast DHFR PCA detects direct and near direct interactions without disturbing endogenous regulation, giving insight into the role of transcriptional regulation in the evolution of PPIs (Tarassov et al., 2008; Rochette et al., 2014; Barshir et al., 2018; Gagnon-Arsenault et al., 2013). PCA is one of the standard binary methods used to measure direct and near-direct PPIs in yeast and mammalian cells (Titeca et al., 2019). PCA’s performance compares to other standard methods when proper controls and analyses are performed. It has been used successfully by our group and others in various contexts since its first application (Schlecht et al., 2017; Celaj et al., 2017; Chrétien et al., 2018; Stynen et al., 2018; Lev, Volpe and Ben-Aroya 2014).

In general, the PCA signal in our study strongly correlates with results from previous PCA experiments (Stynen et al., 2018; Tarassov et al., 2008) and other publicly available data (Figure 2—figure supplement 1). Roughly 75% of the HMs and HETs detected in our PCA experiments were previously reported (Figure 2—figure supplement 2, Tables S3 and S4), suggesting that most of the HMs and HETs that can be detected with the available tools and in standard conditions have been discovered. While 76 HMs and 47 HETs reported in other studies were not detected in our PCA, our experiments detected 44 HMs and 19 HETs not previously reported (Tables S3 and S4).”

4) We simplified the combined the third and fourth paragraphs of subsection “Paralogous heteromers frequently derive from ancestral homomers”. The new paragraph is as follows:

“We classified paralog pairs into four classes according to whether they show only the HET (HET, 10%), at least one HM but no HET (HM, 39%), at least one of the HM and the HET (HM&HET, 37%) or no interaction (NI, 15%) (Figure 2. D, supplementary text). Overall, most pairs forming HETs also form at least one HM (79%, Table S3). For the rest of the study, we focused our analysis and comparisons on HM and HM&HET pairs because they most likely derive from an ancestral HM. Previous observations showed that paralogs are enriched in protein complexes comprising more than two distinct subunits, partly because complexes evolved by the initial establishment of self-interactions followed by duplication of homomeric proteins (Musso et al., 2007; Pereira-Leal et al., 2007). However, we find that the majority of HM&HETs could be simple oligomers of paralogs that do not involve other proteins and are thus not part of large complexes. Only 70 (41%) of the 169 cases of HM&HET are in complexes with more than two distinct subunits among a set of 5,535 complexes reported in databases (see methods).”

5) We combined panels A, B, C and D of Figure 3 into one, and panels B, D E of Figure 6 into only one:

6) We removed the paragraph in the Discussion section that explained how dependent paralogs often form HMs because this is largely speculative:

“Our simulations are consistent with the compensatory model where some pairs of mutations in the two subunits of the HET have opposite effects on binding energy. On the long term, the accumulation of opposite effect mutations could maintain the HET and it could become the only functional unit capable of performing the ancestral function. However, our data suggests that most (89%) of the dependent paralogs that form HET in (Diss et al., 2017) also form at least one HM, suggesting that the loss of both HMs is not required for dependency. Further experiments will be needed to fully determine the likelihood of the dependency model and in which conditions it could take place.”

6) Second paragraph of Results section discusses the effect of expression on detecting HMs by PCA. Since expression has an effect on detection of PPIs, is the difference of HMs among singletons, SSDs and WGDs (mentioned in subsection “Homomers and heteromers in the yeast PPI network”) due to this reason? Would it be feasible for the authors to collect subsets of singletons, SSDs and WGDs with similar magnitudes of expression (use bins) and check difference of HMs for these 3 controlled subsets?

Indeed, the higher expression level of duplicated genes compared to singletons could have an impact in the observed difference of HMs. We controlled the expression level as a factor using a GLM (Supplementary file 2—table S7A) and showed that both factors, expression and duplication, have significant effects on the probability of proteins to form HMs. However, in the first submission, the paragraph discussing the effect of the expression on HM detection was separated (after Figure 2) from the previous paragraph on the proportions of HM. This could be confusing so we grouped the two paragraphs together (subsection “Homomers among singletons and paralogs in the yeast PPI network”):

“Another explanation is that proteins forming HMs could be expressed at higher levels and therefore, easier to detect, as shown above. High expression could also itself increase the long term probability of genes to persist after duplication (Gout et al., 2010; Gout and Lynch, 2015). We observed that both SSDs and WGDs are more expressed than singletons at the mRNA and protein levels, with WGDs being more expressed than SSDs at the mRNA level (Figure 2—figure supplement 4A-B). However, expression level does not explain completely the enrichment of HMs among duplicated proteins and the enrichment does not result entirely from enhanced detection sensitivity. Both factors, expression and duplication, have significant effects on the probability of proteins to form HMs (Supplementary file 2—table S7A). It is therefore likely that the overrepresentation of HMs among paralogs is linked to their higher expression but other factors are also involved.”

7) Please clarify statistics in Supplementary file 2—table S5. More information should be included in that worksheet or somewhere else.

We clarified information in the Supplementary file 2—table S7 (Table S5 in the original submission) descriptions (see the Supplementary material section), added details about the factors tested and data sources on the right side of each table and added the number of observations (N); Akaike Information Criterion (AIC); Bayesian Information Criterion (BIC) and pseudo regression (Pseudo R2).

8) In Figure 2F, although there are some statistically significant differences, the various groups span similar orders of magnitude. Please comment on this observation.

In the previous Figure 2F, HM and HM&HET of SSDs showed significant differences in pairwise amino acid sequence identity. After further filtering the sets of paralogs (see answer 2), the difference is only marginally significant (P=0.065). However, we still observed a wider distribution for WGDs, which at this point is at least partly caused by the two distinct origins of WGDs. We replaced the boxplots by violin plots to be able to distinguish the distributions. This new figure shows that the distribution is different between HM and HM&HET WGD pairs.

We added a comment in the manuscript (subsection “Paralogous heteromers frequently derive from ancestral homomers”):

‘Higher protein sequence divergence could lead to the loss of HET complexes because it increases the chance of divergence at the binding interface. We indeed found that among SSDs, those forming HM&HET tend to show a marginally higher overall sequence identity (p=0.065, Figure 2F, Figure 2—figure supplement 5B and C). We also observed a significantly higher sequence identity for WGD pairs forming HM&HET, albeit with a wider distribution (Figure 2F, Figure 2—figure supplement 5B,C). This wider distribution at least partly derives from the mixed origin of WGDs (Figure 2—figure supplement 5D and E). Recently, Marcet-Houben and Gabaldón (MarcetHouben and Gabaldón, 2015; Wolfe, 2015) showed that WGDs likely have two distinct origins: actual duplication (generating true ohnologs) and hybridization between species (generating homeologs).’

9) Optional: Is it feasible for the authors to do an extra series of wet-lab experiments and experimentally test the HMs and HETs of a selected protein with crystal structure that underwent simulated evolution with the different selection scenarios? That would strengthen the paper further.

We thank the reviewers for this interesting proposal. We are currently running mutagenesis experiments to test the effects of mutations on protein-protein interactions by PCA but it is not feasible to achieve this within the time allocated for revisions.